# NAD kinase promotes *Staphylococcus aureus* pathogenesis by supporting production of virulence factors and protective enzymes

**Clarisse Leseigneur[1], Laurent Boucontet[2], Magalie Duchateau[3], Javier Pizarro-Cerda[1], Mariette Matondo[3], Emma Colucci-Guyon[2], Olivier Dussurget[1]***

[1]Institut Pasteur, Université Paris Cité, CNRS UMR6047, Unité de Recherche Yersinia, Paris, France; [2]Institut Pasteur, Université Paris Cité, CNRS UMR3738, Unité Macrophages et Développement de l'Immunité, Paris, France; [3]Institut Pasteur, Université Paris Cité, CNRS USR2000, Unité de Spectrométrie de Masse pour la Biologie, Plateforme de protéomique, Paris, France

**\*For correspondence:**
olivier.dussurget@pasteur.fr

**Competing interest:** The authors declare that no competing interests exist.

**Abstract** Nicotinamide adenine dinucleotide phosphate (NADPH) is the primary electron donor for reductive reactions that are essential for the biosynthesis of major cell components in all organisms. Nicotinamide adenine dinucleotide kinase (NADK) is the only enzyme that catalyzes the synthesis of NADP(H) from NAD(H). While the enzymatic properties and physiological functions of NADK have been thoroughly studied, the role of NADK in bacterial pathogenesis remains unknown. Here, we used CRISPR interference to knock down NADK gene expression to address the role of this enzyme in *Staphylococcus aureus* pathogenic potential. We find that NADK inhibition drastically decreases mortality of zebrafish infected with *S. aureus*. Furthermore, we show that NADK promotes *S. aureus* survival in infected macrophages by protecting bacteria from antimicrobial defense mechanisms. Proteome-wide data analysis revealed that production of major virulence-associated factors is sustained by NADK. We demonstrate that NADK is required for expression of the quorum-sensing response regulator AgrA, which controls critical *S. aureus* virulence determinants. These findings support a key role for NADK in bacteria survival within innate immune cells and the host during infection.

## Editor's evaluation

This work demonstrates that nicotinamide adenine dinucleotide phosphate is essential for the virulence of *Staphylococcus aureus*. The enzyme that catalyses the final step in the synthesis of this cofactor promotes virulence by protecting bacteria from host antimicrobial defences and is also involved in quorum sensing mechanisms. These data shed new light on the pathogenesis of *S. aureus*.

## Introduction

Nicotinamide adenine dinucleotide (NAD+), its phosphorylated form (NADP+) and their reduced equivalents (NADH and NADPH) are essential cofactors shared by all living organisms, including bacteria. While NAD+ and NADH are important for cellular energy metabolism, inflammation, and senescence (*Chini et al., 2021*; *Covarrubias et al., 2020*), NADP+ and NADPH are key cofactors in central

metabolism, being involved in tricarboxylic acid (TCA) cycle, pentose phosphate pathway as well as de novo synthesis of fatty acids, cholesterol, amino acids and nucleotides (*Chandel, 2021*). NADPH also provides the reducing power necessary for the restoration of antioxidative defense systems of the cell (*Chandel, 2021*). Like NAD(H), growing evidence suggests that NADP(H) has a broader role. In particular, the NADP$^+$ derivative nicotinic acid adenine dinucleotide phosphate (NAADP), which is the major intracellular calcium mobilizing molecule, links NADP(H) metabolism with calcium homeostasis and signaling, development, and differentiation (*Galione and Chuang, 2020*).

Whereas there are two known multi-step pathways for NAD$^+$ biosynthesis, a single enzyme is responsible for the phosphorylation of NAD$^+$/NADH into NADP$^+$/NADPH: the NAD kinase (NADK) (*McGuinness and Butler, 1985*; *Chini et al., 2021*). NADK activity was first reported in the late 30's (*Vestin, 1937*; *Von Euler and Adler, 1938*), and the enzyme was purified from yeast by *Kornberg, 1950*. It is only in 2000 that the genes encoding NADK were identified in *Micrococcus flavus* and *Mycobacterium tuberculosis* (*Kawai et al., 2000*). Since then, NADK genes have been identified in all living organisms, except the intracellular parasitic bacteria, *Chlamydia* spp. on the basis of genome annotation (*Grose et al., 2006*; *Fisher et al., 2013*). Given the vital role of NADPH, notably during oxidative stress, NADK genes have been shown to be essential for growth of several bacteria, such as *Mycobacterium tuberculosis* (*Sassetti et al., 2003*), *Bacillus subtilis* (*Kobayashi et al., 2003*), *Salmonella enterica* (*Grose et al., 2006*), and *Staphylococcus aureus* (*Chaudhuri et al., 2009*; *Gelin et al., 2020*). NADKs essentiality may account for the poor characterization of their role in prokaryotes. In contrast, NADKs have been extensively studied in plants (*Li et al., 2018*), especially in *Arabidopsis thaliana* which possesses three NADK encoding genes (*Chai et al., 2006*; *Turner et al., 2004*). NADK has a key role in photosynthesis, plant cell metabolism, intracellular redox balance (*Li et al., 2018*), and response to stresses as shown during exposure to aluminum in wheat (*Ślaski, 1995*), upon cold shock in green bean leaves (*Ruiz et al., 2002*), upon treatment with NaCl, ionizing radiations or oxidative stress in *Arabidopsis thaliana* (*Chai et al., 2006*; *Berrin et al., 2005*). In plants, NADKs are regulated by calcium and calmodulin (*Tai et al., 2019*). In contrast, in prokaryotes allosteric regulation of NADK activity by NADPH or NADH is the main mechanism identified so far (*Grose et al., 2006*).

The importance of microbial metabolic adaptation during infection (*Eisenreich et al., 2010*; *Richardson, 2019*; *Teoh et al., 2021*) and the unknown contribution of NADK to bacterial pathogenesis prompted us to investigate the role of NADK in *Staphylococcus aureus* pathogeny. While being carried as a commensal by around one-third of the human population, *S. aureus* is a leading cause of infectious diseases, ranging from mild skin and soft tissue infections to life-threatening endocarditis and bacteremia (*Turner et al., 2019*). Worryingly, *S. aureus* can develop resistance to virtually all antibiotic classes available (*Vestergaard et al., 2019*), limiting dramatically therapeutic options for some patients. Therefore, WHO included *S. aureus* in the list of high-priority pathogens to promote research and development of new antibiotics and emphasized the need for innovation and diversification in the choice of targets (https://www.who.int/news/item/27-02-2017-who-publishes-list-of-bacteria-for-which-new-antibiotics-are-urgently-needed).

In this study, we used a genetic approach to modulate NADK activity and analyze *S. aureus* behavior in the infected host, in infected cells and in response to stresses mimicking those encountered during infection. We establish the first link between NADK activity and *S. aureus* pathogenic potential, as NADK inhibition led to an increased survival of the host in a zebrafish infection model. We further demonstrate that inhibition of NADK leads to a dramatic decrease of *S. aureus* survival in macrophages, highlighting the enzyme importance during interactions with immune cells. We further show that exposures to oxidative and envelope-targeting antimicrobial compounds significantly decrease bacterial survival upon inhibition of NADK activity. Proteomic analysis indicated that the production of protective enzymes, virulence factors and the quorum-sensing response regulator AgrA relies on NADK. Thus, our results reveal a key role for NADK in bacterial pathogenesis.

## Results

### NADK contributes to *S. aureus* virulence in zebrafish

As most prokaryotes, *S. aureus* possesses a single NADK encoded by the highly conserved *ppnK* gene. To investigate the role of NADK in *S. aureus* pathogenesis, we followed a genetic approach to modulate the expression of *ppnK*. We used the *S. aureus* NADK sgRNA strain (*Gelin et al., 2020*), in which

the levels of *ppnK* expression were decreased by CRISPR interference based on plasmid pSD1 (*Zhao et al., 2017*), without affecting the transcript levels of the downstream gene *rluA* (*Figure 1—figure supplement 1*). Knockdown of NADK expression in this strain was analyzed by RT-PCR, immunoblotting, and aerobic growth in BHI at 37°C (*Figure 1—figure supplement 1*). Growth of the *S. aureus*/sgRNA strain was affected as previously reported (*Gelin et al., 2020*), attesting the efficiency of the knockdown. We investigated whether NADK inhibition might influence *S. aureus* pathogenesis in a zebrafish infection model. This model allows short kinetics of infection (*Prajsnar et al., 2008*) and has been previously used to study *S. aureus* infection dynamics (*Prajsnar et al., 2012*). In addition to being optically transparent, zebrafish larvae innate immune system is the only one operating during the first days of development, facilitating the study of interactions between bacteria and neutrophils or macrophages. We first determined the stability of the pSD1 plasmid during infection. Zebrafish larvae were infected at 60 hr post fertilization (hpf) by intravenous injections of either *S. aureus* containing the empty vector pSD1 or the knockdown NADK sgRNA strain. Bacteria recovered from infected larvae were plated onto BHI agar with and without chloramphenicol. There was no significant difference between the number of bacteria growing on BHI alone and BHI supplemented with chloramphenicol up to 48 hr post-infection (hpi), irrespective of the strain (*Figure 1—figure supplement 2A*). Thus, plasmids were maintained in our experimental infection conditions. We next monitored zebrafish survival upon infection with $10^4$ bacteria. *S. aureus*/pSD1 infection led to ≈ 75% fish mortality at 24 hpi (*Figure 1A*). Strikingly, only one larva out of 48 died when bacterial NADK was inhibited (*Figure 1A*). Decreased fish mortality upon *S. aureus*/NADK sgRNA infection was also observed 48 and 72 hpi (*Figure 1A*). These results indicate that NADK inhibition leads to a decreased bacterial virulence. We then determined the bacterial burden of fish larvae 24 and 48 hpi. Interestingly, numbers of *S. aureus*/pSD1 and *S. aureus*/NADK sgRNA bacteria at both time points varied depending on the viability of the fish (*Figure 1B*). As observed previously by *Prajsnar et al., 2012*, some fishes were able to control the infection with $10^4$ bacteria but a high bacterial burden was associated with larval death in most cases (*Figure 1B*). Early mortality was observed in zebrafish in which *S. aureus*/pSD1 developed to high numbers compared to larvae infected with the NADK knockdown strain. Together, these results suggest that NADK contributes to *S. aureus* pathogenic potential, which depends on both bacterial and host factors. To investigate the impact of NADK inhibition on phagocyte populations, we infected intravenously transgenic zebrafish lines with *S. aureus*/pSD1 or *S. aureus*/NADK sgRNA strains, and observed bacteria, macrophages, and neutrophils at time 0, 6, 12, and 24 hpi. *S. aureus*/pSD1 bacterial burden was higher than that of *S. aureus*/NADK sgRNA strain in head and caudal regions observed in fish of comparable viability 12 hpi (*Figure 1C*) and 24 hpi (*Figure 1—figure supplement 2B*). We also noticed a strong depletion of neutrophils in larvae infected with *S. aureus*/pSD1, while this neutropenia was more limited upon infection with NADK knockdown bacteria (*Figure 1C, D and E*). This correlated with virulence differences of the two strains, given the key role of neutrophils in *S. aureus* infection (*Pollitt et al., 2018*; *Prajsnar et al., 2012*). In contrast, the number of macrophages in larvae did not significantly decrease upon infection with either strain (*Figure 1—figure supplement 2C*). Altogether, our results suggest that NADK is important for *S. aureus* pathogenesis.

## NADK promotes *S. aureus* survival in macrophages

Since both neutrophils and macrophages are key immune cells specialized in the destruction of microorganisms, we next investigated if, in addition to triggering neutrophil killing, NADK might support *S. aureus* survival during macrophages infection. Murine RAW264.7 macrophages were infected with *S. aureus*/pSD1 or *S. aureus*/NADK sgRNA strains. Bacterial enumerations were performed at time zero and at 1, 2, 4, and 6 hpi. No significant difference in initial phagocytosis of the two strains could be observed, nor at early time of infection (*Figure 2A*). In contrast, after 2 hr of infection the percentage of intracellular bacteria was significantly reduced upon *ppnK* knockdown, and the survival defect increased over time (*Figure 2A*), indicating that bacteria were not able to survive inside macrophages without proper NADK activity. As oxidative burst is a major antimicrobial mechanism of phagocytes (*Fang, 2004*), we next investigated the role of ROS produced by macrophages in the control of the *S. aureus*/NADK sgRNA strain using the antioxidant agent N-acetyl cysteine (NAC). NAC partially restored bacterial survival (*Figure 2A*), indicating that the bacterial growth defect of the *ppnK* knockdown strain was due to its inability to cope with oxidative stress to some extent. Using confocal fluorescence microscopy, we then investigated bacterial intracellular localization in macrophages 6hpi.

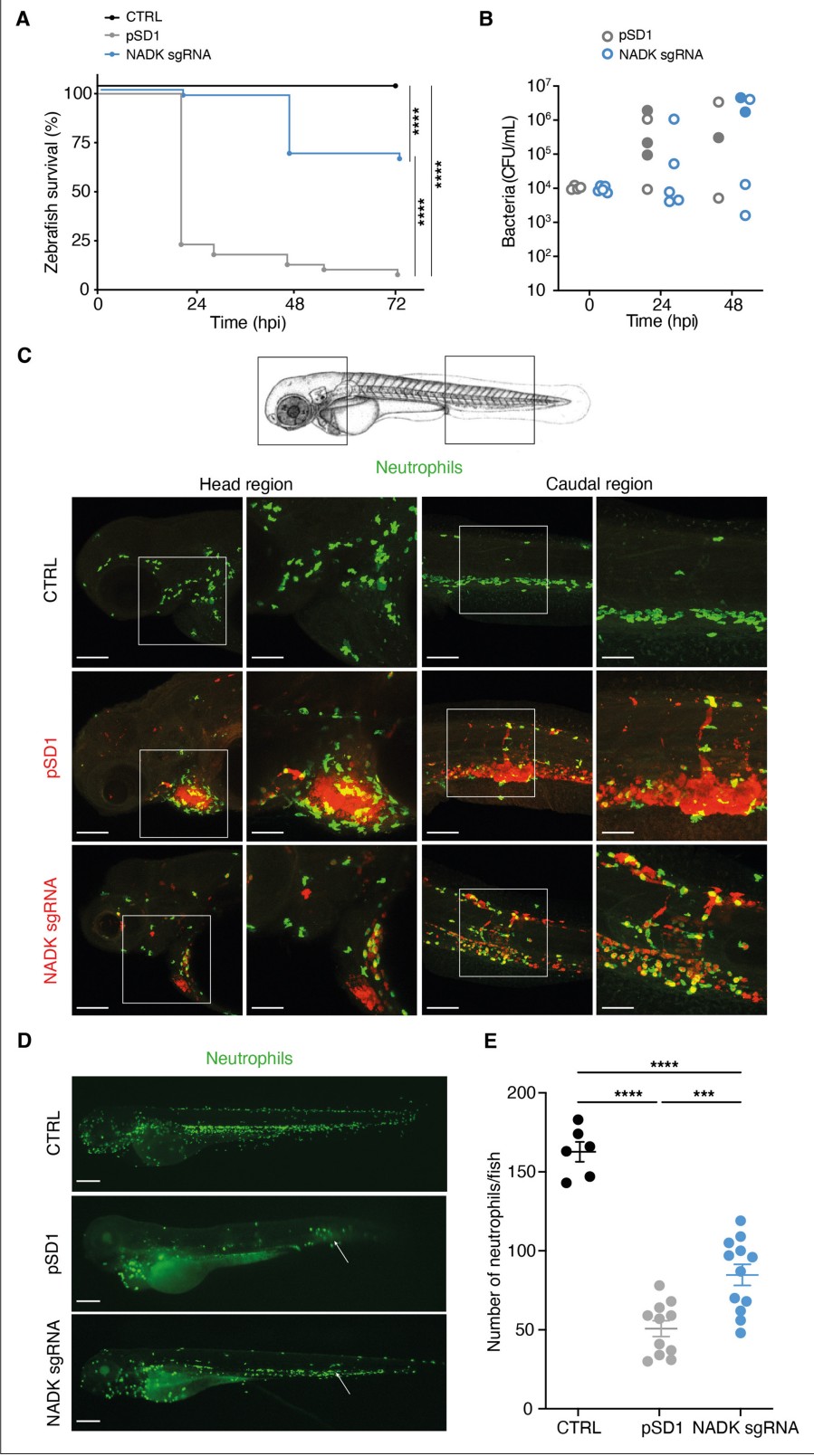

**Figure 1.** Nicotinamide adenine dinucleotide kinase (NADK) promotes *S. aureus* virulence in zebrafish. (**A**) Survival of zebrafish larvae uninfected (CTRL) or intravenously injected at 60 hpf with $10^4$ *S. aureus* containing the empty vector (pSD1) or the NADK knockdown strain (NADK sgRNA) between 0 and 72 hpi (n=48). (**B**) Bacterial burden in zebrafish larvae upon intravenous injection with $10^4$ *S. aureus* containing the empty vector (pSD1) or the NADK

*Figure 1 continued*

knockdown strain (NADK sgRNA). For each strain, CFU was determined in living larvae (open circles) or dead larvae (filled circles) 0, 24, and 48 hpi. (**C**) Representative fluorescence confocal images of transgenic *mpx:GFP* zebrafish larvae uninfected (CTRL), or intravenously injected with $10^4$ *S. aureus* containing the empty vector (pSD1) or the NADK knockdown strain (NADK sgRNA) at 12 hpi. Maximum intensity Z-projection images (2 μm serial optical sections) of bacteria (red) and neutrophils (green). Scale bars, 25 μm. Insets are shown at higher magnification on the right panels for head and caudal regions. Scale bars, 10 μm. Neutrophils containing NADK knockdown bacteria can be seen in the caudal inset. (**D**) Representative fluorescence confocal images of transgenic *mpx:GFP* zebrafish larvae uninfected (CTRL), or intravenously injected with $10^4$ *S. aureus* containing the empty vector (pSD1) or the NADK knockdown strain (NADK sgRNA) at 24 hpi, showing neutrophils (green). White arrows indicate the injection site. Scale bars, 500 μm (**E**) Number of neutrophils in uninfected zebrafish larvae (CTRL) or in zebrafish larvae intravenously injected with $10^4$ *S. aureus* containing the empty vector (pSD1) or the NADK knockdown strain (NADK sgRNA) at 24 hpi. Comparison of data was performed using one-way analysis of variance (***$p<0.001$, ****$p<0.0001$).

The online version of this article includes the following source data and figure supplement(s) for figure 1:

**Figure supplement 1.** Nicotinamide adenine dinucleotide kinase (NADK) is important for *S. aureus* growth.

**Figure supplement 1—source data 1.** NADK transcript levels (A).

**Figure supplement 1—source data 2.** NADK protein levels (B).

**Figure supplement 1—source data 3.** NADK operon transcript levels (E).

**Figure supplement 1—source data 4.** RluA transcript levels (F).

**Figure supplement 2.** Nicotinamide adenine dinucleotide kinase (NADK) contributes to *S. aureus* virulence in zebrafish.

Few *S. aureus*/NADK sgRNA bacteria were detected compared to *S. aureus*/pSD1 (**Figure 2B**), and NAC treatment led to an increased number of bacteria (**Figure 2C**), confirming bacterial enumeration results. Interestingly, while all knockdown bacteria colocalized with lysosomal associated membrane protein 1 (LAMP1), many *S. aureus*/pSD1 did not upon NAC treatment, suggesting that NADK activity might be required for phagolysosome escape and bacterial survival in macrophages. Since *S. aureus* has been shown to be cytotoxic to macrophages, we next wondered if the bacterial growth defect of the NADK knockdown strain was correlated with reduced lysis of infected cells. We measured the levels of cell death in macrophages that were uninfected, uninfected, and treated with Triton X-100 or infected with *S. aureus*/pSD1 or *S. aureus*/NADK sgRNA strains (**Figure 2D**). Detergent treatment expectedly led to massive cell lysis. Infections produced significant macrophage death compared to uninfected cells, although to much lower levels than that triggered by Triton X-100. Infection with the *ppnK* knockdown strain was significantly less cytotoxic than infection with the *S. aureus*/pSD1 strain. Altogether, these results suggest that NADK promotes survival and possibly virulence of *S. aureus* in macrophages.

## NADK protects *S. aureus* from antimicrobial defense compounds

Hydrogen peroxide is one of the reactive oxygen species generated upon activation of macrophages (**Fang, 2004**). We, therefore, compared growth of the *S. aureus* strain containing the empty vector to that of the NADK knockdown strain, upon exposure to increasing concentrations of hydrogen peroxide ($H_2O_2$). While the *S. aureus*/pSD1 strain resisted to high doses of $H_2O_2$, the *S. aureus*/NADK sgRNA strain showed increasing growth deficiency when exposed to $H_2O_2$ concentrations ranging from 50 to 500 μM $H_2O_2$ (**Figure 3A**). Expectedly, treatment of the *S. aureus*/pSD1 strain with catalase alone or catalase and $H_2O_2$ did not affect growth (**Figure 3B**). In contrast, addition of catalase, which catalyzes hydrogen peroxide dismutation into water and oxygen, rescued the growth defect of the *S. aureus*/NADK sgRNA strain upon $H_2O_2$ treatment (**Figure 3B**). Taken together, our results show that NADK protects *S. aureus* from toxic effects of a reactive oxygen species involved in host defense. Within phagocytes, bacteria have to cope with other mechanisms of antimicrobial defenses such as production of lysozyme (**Callewaert and Michiels, 2010**; **Avican et al., 2021**; **Fang et al., 2016**). We thus investigated the effects of NADK knockdown on *S. aureus* sensitivity to lysozyme. Since *S. aureus* is highly resistant to lysozyme, we treated bacteria with lysozyme and the bacteriocin lysostaphin, a combination that potentiates their antibacterial activity against *S. aureus* (**Cisani et al., 1982**). At subinhibitory concentrations for the *S. aureus*/pSD1 strain, the *S. aureus*/NADK sgRNA strain showed

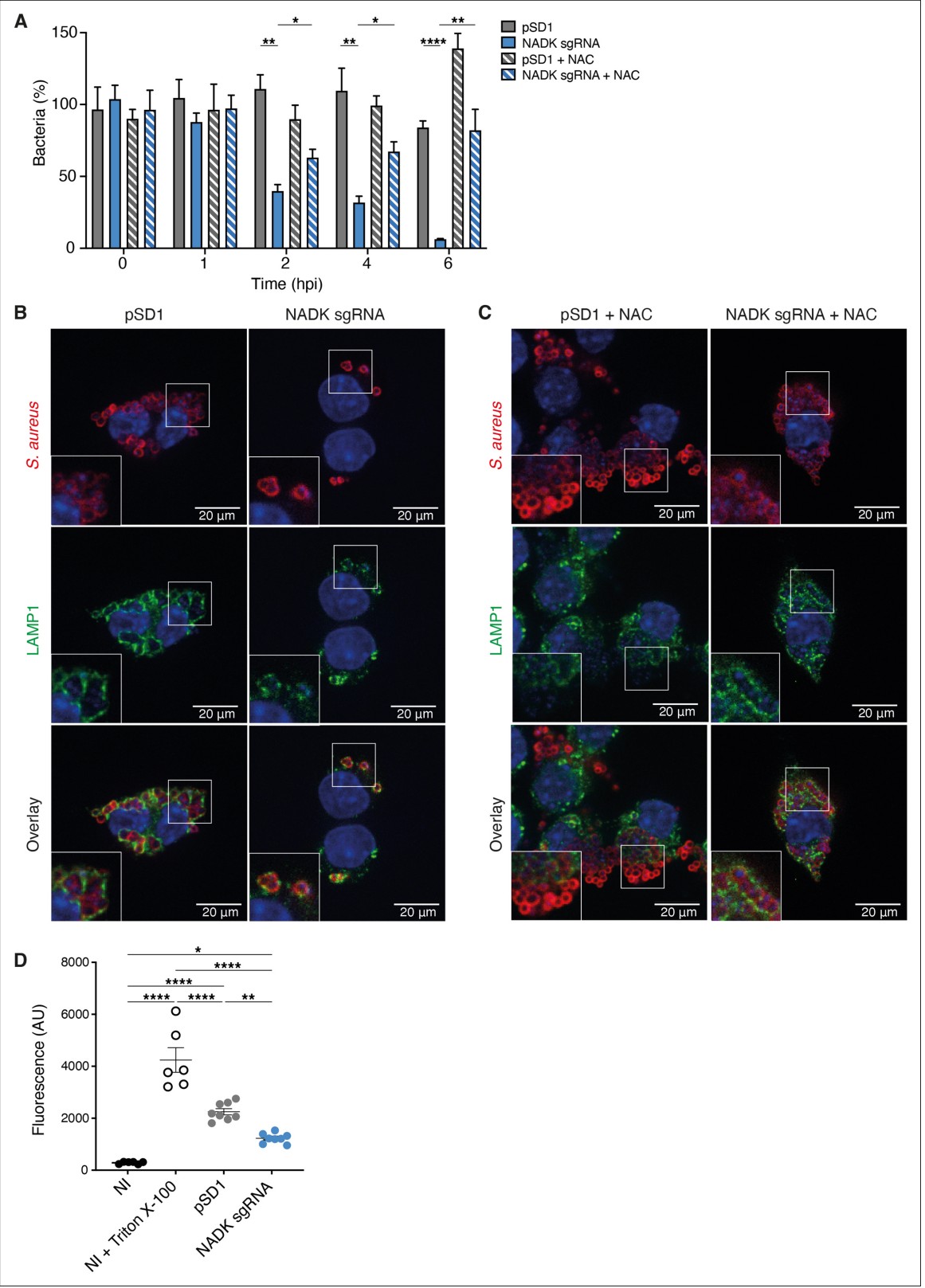

**Figure 2.** Nicotinamide adenine dinucleotide kinase (NADK) promotes *S. aureus* survival and cytotoxicity in macrophages. (**A**) Percentage of growth of the *S. aureus* strains containing the empty vector (pSD1) or the *ppnK* knockdown strain (NADK sgRNA) at time 0, and 1, 2, 4, and 6 hpi of RAW264.7 macrophages left untreated or treated with N-acetylcysteine (NAC). Bars indicate standard errors of the means of biological replicates (n=4). Comparison of data was performed using two-ways analysis of variance (*p<0.05, **p<0.01, ****p<0.0001). (**B–C**) Representative images of RAW264.7

*Figure 2 continued on next page*

*Figure 2 continued*

macrophages infected with *S. aureus*/pSD1 or *S. aureus*/NADK sgRNA strains and analyzed 6 hpi by confocal fluorescence microscopy using antibodies to label *S. aureus* (Cy5, red) and LAMP-1 (FITC, green). Nuclei were labeled with DAPI (blue). (**B**) shows untreated macrophages. (**C**) shows macrophages treated with NAC. Insets are shown at higher magnification. Images are representative of three independent experiments. (**D**) Cell death of RAW264.7 macrophages uninfected (NI), uninfected and treated with Triton X-100 (lysis control), or 6 hpi with *S. aureus* strains containing the empty vector (pSD1) or the *ppnK* knockdown strain (NADK sgRNA). Bars indicate standard errors of the means of biological replicates. Comparison of data was performed using two-ways analysis of variance (*p<0.05, **p<0.01, ****p<0.0001). Data are representative of at least three independent experiments.

decreased viability when exposed to lysozyme and lysostaphin (*Figure 3C*). Thus, NADK protects *S. aureus* from toxic effects of cell envelope-targeting compounds.

## NADK supports *S. aureus* protective enzymes

To elucidate mechanisms of action of NADK, unbiased mass spectrometry-based proteomics was used to identify proteins differentially expressed upon NADK knockdown. Relative protein abundance was compared in the *S. aureus* strain containing the empty vector and the NADK knockdown strain grown in BHI. Interestingly, lower levels of major antioxidant enzymes were detected in the NADK sgRNA strain (*Table 1* and *Supplementary file 1*). NADK deficiency was linked to decreased relative abundance of superoxide dismutase SodM, catalase KatA, thioredoxin Trx2 and methionine sulfoxide reductase MsrA2. Additionally, the NADK sgRNA strain had lower levels of the peptidoglycan *O*-acetyltransferase OatA and the glycopeptide resistance-associated two-component system GraRS, major determinants for *S. aureus* resistance to lysozyme and cationic antimicrobial peptides (CAMP), respectively (*Bera et al., 2005*; *Herbert et al., 2007*). We then compared the proteomes of the *S. aureus* strain containing the empty vector and the NADK knockdown strain upon hydrogen peroxide treatment. OatA, SodM, Trx2, MrsA2, KatA, and other antioxidant enzymes were more abundant in the *S. aureus*/pSD1 strain than in the NADK deficient strain (*Supplementary files 2 and 3*), as it had been observed when strains were grown in BHI. To determine the effect of NADK knockdown on *S. aureus* proteome in vivo, relative protein abundance was analyzed in zebrafish infected with the *S. aureus* strain containing the empty vector or the NADK knockdown strain. The D-alanyl carrier protein DltC, which mediates CAMP resistance, and antioxidant enzymes, such as SodA, KatA, thiol peroxidase Tpx, and alkyl hydroperoxide reductase subunit AhpF, were detected in zebrafish infected with *S. aureus*/pSD1 (*Supplementary files 4 and 5*). In contrast, these proteins were undetectable in *S. aureus*/NADK sgRNA-infected zebrafish. Thus, our approach identified a role for NADK in supporting *S. aureus* protective proteins in vitro and in vivo.

## NADK promotes virulence factor expression by *S. aureus*

To investigate additional mechanisms of action of NADK, our quantitative proteomic data were analyzed further using proteomaps (*Liebermeister et al., 2014*). *S. aureus* proteins were classified into functional groups based on KEGG annotation. Expression profiles of strains grown in BHI were visualized by Voronoi tree maps, clustering proteins in three hierarchical categories. Top-level functional categories relying on NADK activity included biosynthesis, central carbon metabolism, other enzymes, folding, sorting and degradation, translation, transcription, and DNA maintenance (*Figure 4A*). Remarkably, infectious diseases, signaling molecules and interaction, and signal transduction identified at the highest level (*Figure 4A*), and *S. aureus* infection, bacterial toxins, and two-component system identified at the second level (*Figure 4B*), encompassed multiple major virulence factors such as alpha-hemolysin, gamma-hemolysin, Map, IsdA, and virulence regulators AgrA and SaeS (*Figure 4C*). NADK deficiency was also linked to decreased relative abundance of other virulence-related proteins, for example, phenol-soluble modulin (PSM) alpha PsmA, the lipase Lip1, leukocidin-like proteins 1 and 2, and staphopain A (*Table 2*). Of note, similar proteomic profiles were obtained when strains were exposed to hydrogen peroxide, confirming the importance of NADK for proper expression of virulence proteins in conditions mimicking host defenses (*Figure 4—figure supplement 1*, *Supplementary file 6*). Furthermore, comparative analysis of whole protein extracts from bacterial pellets and culture supernatants by immunoblotting showed a massive decrease in the levels of alpha-hemolysin secretion by the NADK deficient strain (*Figure 4D and E*). To determine if the differential protein abundance of toxins was translated into different levels of enzymatic activity, a Christie-Atkins-Munch-Peterson test was performed using *S. aureus* RN4220, *S. aureus*/pSD1, and

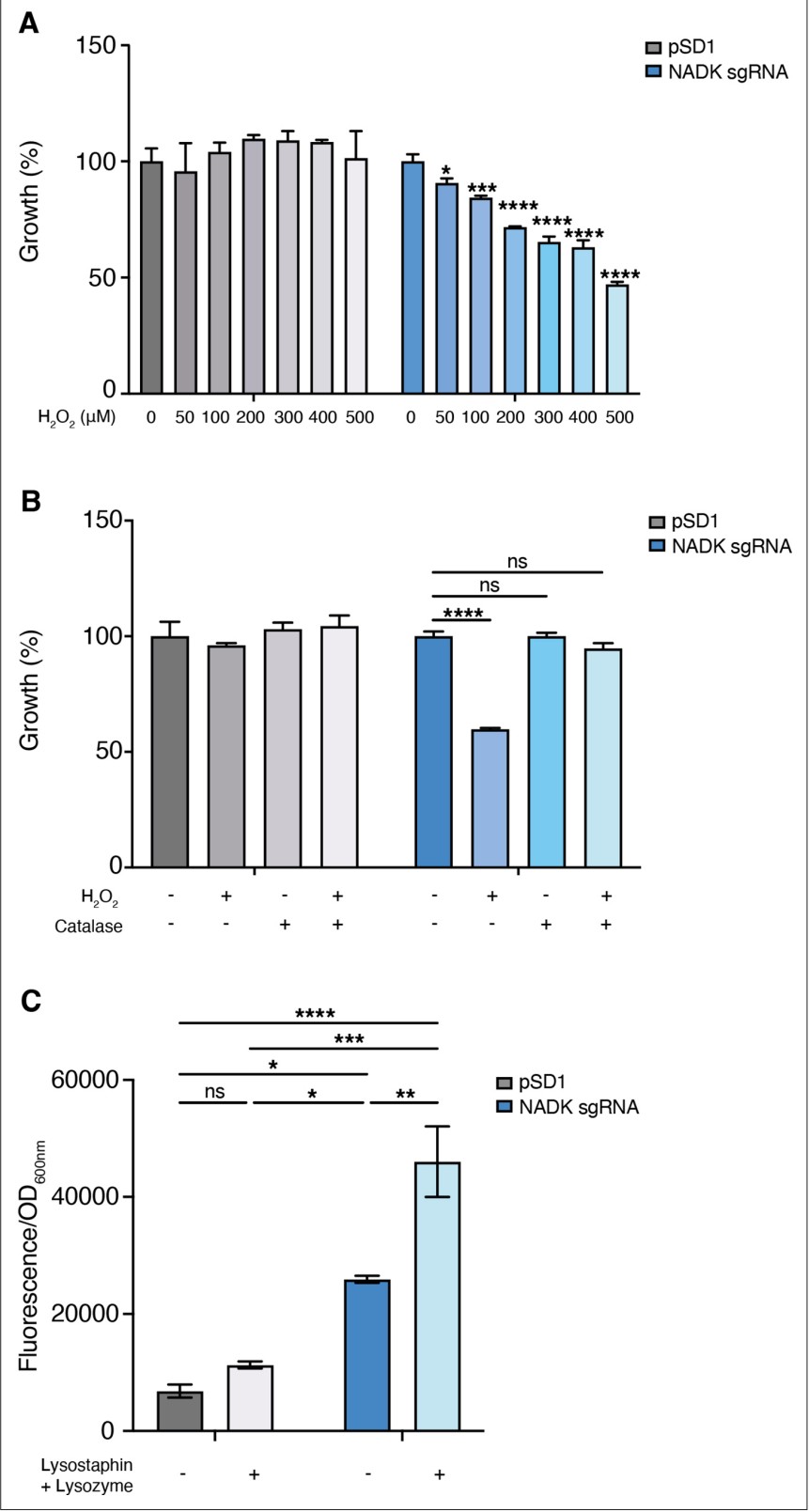

**Figure 3.** Nicotinamide adenine dinucleotide kinase (NADK) protects *S. aureus* from antimicrobial defense compounds. Bacterial growth was monitored at $OD_{600nm}$ in BHI broth at 37°C. (**A**) Percentage of growth of the *S. aureus* strain containing the empty vector (pSD1) and the *ppnK* knockdown strain (NADK sgRNA) exposed for 6 hr to increasing concentrations of $H_2O_2$ relative to the untreated condition. (**B**) Percentage of growth of the *S. aureus*

*Figure 3 continued on next page*

*Figure 3 continued*

strain containing the empty vector (pSD1) and the *ppnK* knockdown strain (NADK sgRNA) exposed for 6 hr to $H_2O_2$ and catalase alone or in combination, relative to the untreated condition. Data shown are representative of three independent experiments. Bars indicate the standard error of the means of biological replicates. Comparison of data was performed using one-way analysis of variance (ns: nonsignificant, *p<0.05, ***p<0.001, ****p<0.0001). (**C**) Bacterial cell death of the *S. aureus* strain containing the empty vector (pSD1) and the *ppnK* knockdown strain (NADK sgRNA) untreated or exposed for 30 min to 0.05 mg/mL lysostaphin and 5 mg/mL lysozyme. Bacterial permeability was assessed using the CellTox Green cytotoxicity assay. Bars indicate standard errors of the means of biological replicates. Comparison of data was performed using two-ways analysis of variance (*p<0.05, **p<0.01, ***p<0.001, ****p<0.0001). Data are representative of at least three independent experiments.

*S. aureus*/NADK sgRNA strains (**Adhikari et al., 2007**). Strikingly, NADK deficiency led to a lack of hemolysis (**Figure 4F**). In contrast *S. aureus*/pSD1 produced a synergistic hemolysis in presence of the RN4220 strain, indicating secretion of hemolysins and phenol soluble modulins. Taken together, these data suggest that NADK is required for expression of *S. aureus* virulence determinants.

## NADK promotes AgrA expression by *S. aureus*

AgrA is a central regulator of *S. aureus* virulon. Therefore, the impact of NADK on AgrA observed above may account for the positive regulation of virulence factor production. To attest NADK-dependent regulation of virulence determinants controlled by Agr, we analyzed transcript levels of two additional members of the Agr regulon, the positively regulated multifunctional RNA RNAIII and the negatively regulated glutathione peroxidase gene *bsaA*. We carried out RT-PCR on RNA purified from the *S. aureus* strain containing the empty vector and the NADK knockdown strain during exponential growth in BHI broth at 37°C. Decreased RNAIII transcript levels and increased *bsaA* transcript levels were detected in the NADK knockdown strain (**Figure 5A, B and D**). These results are in accordance with the positive regulatory role of NADK on AgrA, and the regulation of AgrA on RNAIII and *bsaA* expression. To determine whether NADK sustains AgrA protein levels by promoting gene expression, AgrA transcript levels were compared in the *S. aureus*/pSD1 and *S. aureus*/NADK sgRNA strains. The expression of *agrA* was strongly decreased upon NADK knockdown (**Figure 5C and D**). Thus, full NADK activity is required for optimal *agrA* expression and subsequent virulence gene expression. To decipher the mode of action of NADK on *agrA* expression, we next investigated the effect of the *ppnK* knockdown on *S. aureus* oxidative content, since transcription of the *agr* system is repressed in oxidizing conditions (**Baker et al., 2010**; **Rothfork et al., 2004**). We compared the levels of ROS in the *S. aureus*/pSD1 and NADK sgRNA strains grown in aerobic conditions. Decreasing NADK levels led to a significant increase in ROS content (**Figure 5E**), which could contribute to negative regulation of AgrA expression. Together, our data suggest that NADK activity is important for proper *agrA* expression, possibly by controlling the bacterial redox state, ultimately promoting *S. aureus* virulence potential.

**Table 1.** Relative abundance of a subset of protective proteins differentially expressed by *S. aureus* pSD1 and NADK sgRNA strains.

| Uniprot | Protein | Description | Log2R* | P† |
|---------|---------|-------------|--------|-----|
| Q2FV54 | OatA | O-acetyl transferase | 3.07 | 1.78E-05 |
| Q2G000 | Trx2 | Thioredoxin 2 | 1.79 | 2.75E-05 |
| Q2G261 | SodM | Superoxide dismutase [Mn/Fe] | 1.68 | 2.11E-06 |
| Q2FZZ3 | - | Thioredoxin domain-containing protein | 1.59 | 7.22E-04 |
| Q2G0D9 | GraS | Sensor histidine kinase | 1.31 | 4.00E-04 |
| Q2G280 | - | Peroxiredoxin | 1.26 | 6.69E-07 |
| P0A086 | MsrA2 | Methionine sulfoxide reductase | 1.10 | 2.25E-05 |
| Q2FVL7 | - | Thioredoxin domain-containing protein | 1.02 | 1.94E-03 |
| Q2G0E0 | GraR | Response regulator protein | 1.00 | 2.23E-03 |
| Q2FYU7 | KatA | Catalase | 0.59 | 1.00E-05 |

*Log2$R$=Log2[pSD1]/[NADK sgRNA].
†Adjusted p-value.

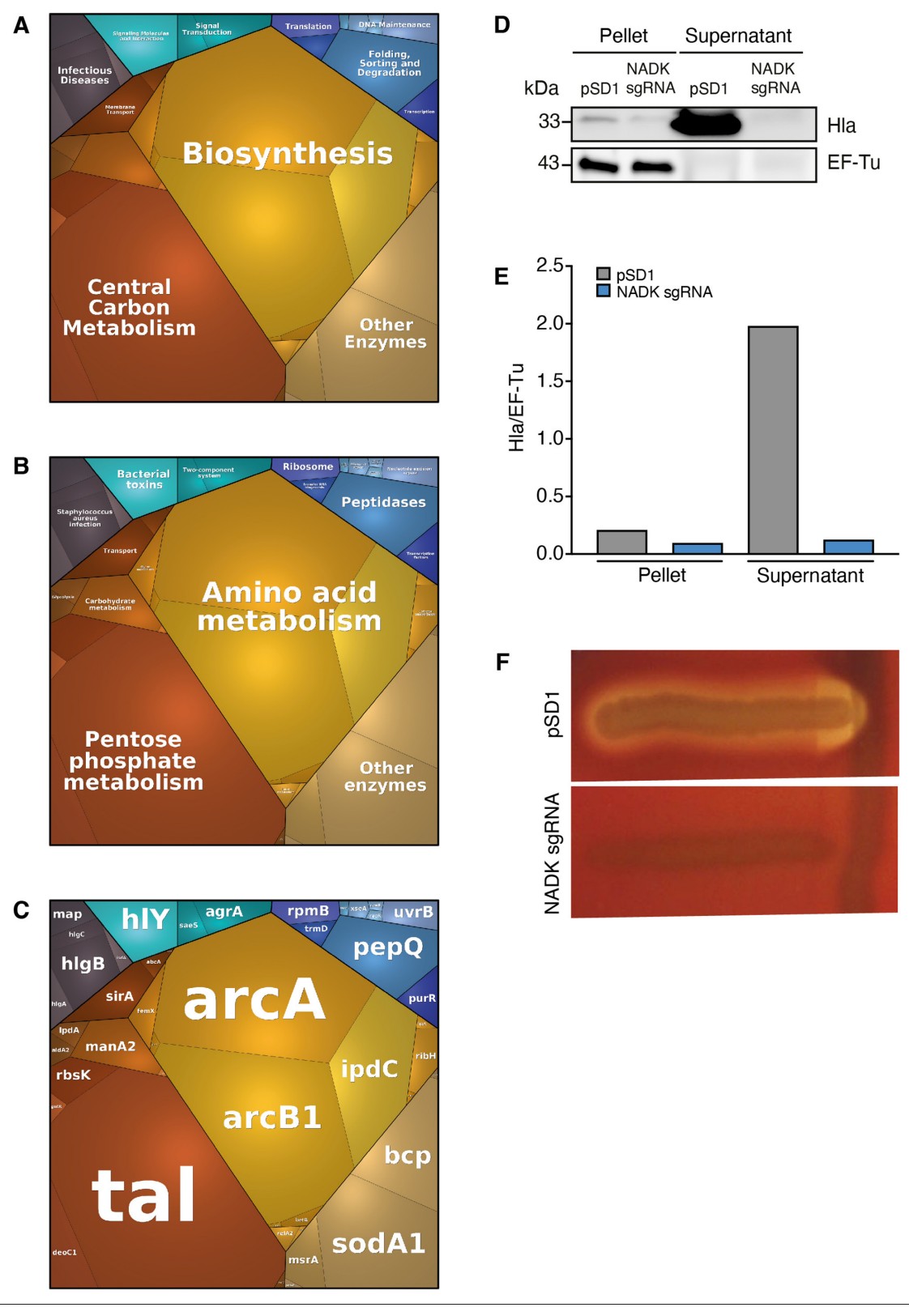

**Figure 4.** Nicotinamide adenine dinucleotide kinase (NADK) promotes expression of *S. aureus* virulence determinants. The *S. aureus* strain containing the empty vector (pSD1) and the *ppnK* knockdown strain (NADK sgRNA) were grown in BHI broth at 37°C. (**A–C**) Whole bacterial cell lysates were analyzed by LC-MS/MS. Voronoi treemaps were generated to visualize proteins more abundant in *S. aureus*/pSD1 than in the knockdown strain at three hierarchical levels according to KEGG database: top level (**A**), second level (**B**), third level (**C**). Colors represent functional categories. Category size

*Figure 4 continued on next page*

*Figure 4 continued*

based on LFQ intensity corresponds to differences in protein abundance. (**D**) Whole bacterial cell lysates and culture supernatants were analyzed by immunoblotting using antibodies against alpha-hemolysin and EF-Tu. (**E**) Quantification of the Hla immunoblots normalized to corresponding EF-Tu protein levels in the pellet. (**F**) A Christie-Atkins-Munch-Peterson test was performed by streaking the *S. aureus* strain containing the empty vector (pSD1) and the *ppnK* knockdown strain (NADK sgRNA) perpendicularly to the *S. aureus* RN4220 strain that produces only beta-hemolysin (vertical streak) on sheep blood agar. Clearing zones indicate hemolysis.

The online version of this article includes the following source data and figure supplement(s) for figure 4:

**Source data 1.** Alpha-hemolysin protein levels (D).

**Source data 2.** Hemolysis levels (F).

**Figure supplement 1.** Nicotinamide adenine dinucleotide kinase (NADK) promotes expression of *S. aureus* virulence determinants.

## Discussion

The ability of *S. aureus* to survive and thrive in a wide variety of niches during infection, ranging from skin to deeper tissues and abiotic devices, mirrors highly plastic metabolic capacities allowing adaptive responses to constantly changing microenvironments (*Potter et al., 2020*). Metabolic pathways are not only necessary for nutrient acquisition to sustain bacterial growth, but also for regulation of virulence and are therefore central to host-pathogen interactions (*Eisenreich et al., 2010*; *Harper et al., 2018*; *Richardson, 2019*; *Teoh et al., 2021*; *Tomlinson et al., 2021*).

Here, we show for the first time that NADK contributes to bacterial pathogenic potential. NADK was necessary for successful *S. aureus* infection of zebrafish larvae, ultimately leading to neutropenia and death. Decreasing NADK activity led to decreased host mortality without affecting bacterial burden initially, suggesting that NADK promotes virulence factor activity and/or counteracts host defenses independently of its contribution to bacterial growth capacity. Besides neutrophils, macrophages are crucial cells in the innate immune defense against infection. We uncovered that inhibition of *S. aureus* NADK activity has a major impact on bacteria interactions with macrophages, reducing dramatically bacterial survival after phagocytosis. We demonstrated that NADK contributes to resistance to ROS produced by macrophages, neutralizing an important defense mechanism against bacterial infection (*Pidwill et al., 2020*). Along the same lines, we showed that NADK protected *S. aureus* from hydrogen peroxide toxicity. NADK, as the only source of NADP(H), is a key component of defense against oxidative stress (*Grose et al., 2006*; *Mailloux et al., 2011*). In bacteria, it has been shown that NADK activity is increased upon exposure to oxidative stress, acting as a metabolic switch to decrease the $NAD^+$ pool which fuels ROS formation and increase the NADPH pool which promotes ROS scavenging activities (*Singh et al., 2007*). Analysis of *S. aureus* proteome revealed a drop in the levels of major antioxidant enzymes upon inhibition of *ppnK* expression. By supporting superoxide dismutase, catalase, methionine sulfoxide reductase, and other antioxidant enzymes, NADK could prevent damages caused by ROS formed during phagocytosis. However, antioxidant treatment of infected macrophages using NAC only partially mitigated growth defect of the NADK knockdown strain. NADK

**Table 2.** Relative abundance of a subset of virulence-related proteins differentially expressed by *S. aureus* pSD1 and NADK sgRNA strains.

| Uniprot | Protein | Description | Log2R* | P† |
|---------|---------|-------------|--------|-----|
| Q2FWM4 | AgrA | Accessory gene regulator protein | + | NA |
| P0C818 | PsmA4 | Phenol-soluble modulin alpha 4 | + | NA |
| Q2FVK2 | HlgC | Gamma-hemolysin component C | + | NA |
| P0C7Y1 | PsmA1 | Phenol-soluble modulin alpha 1 | + | NA |
| Q2FUU5 | Lip1 | Lipase | + | NA |
| Q2FVK1 | HlgB | Gamma-hemolysin component B | 7.88 | 3.85E-10 |
| Q2G1 × 0 | Hly | Alpha-hemolysin | 6.99 | 3.85E-10 |
| Q2FWN9 | LukL2 | Leukocidin-like protein 2 | 3.88 | 4.14E-08 |
| Q2FVK3 | HlgA | Gamma-hemolysin component A | 3.84 | 1.47E-08 |
| Q2FWP0 | LukL1 | Leukocidin-like protein 1 | 3.09 | 2.25E-06 |
| Q2G2R8 | SspP | Staphopain A | 2.89 | 4.08E-07 |

*Log2R=Log2[pSD1]/[NADK sgRNA]; +: protein detected in pSD1 strain and not detected from NADK sgRNA strain.

†Adjusted p-value: NA: not applicable.

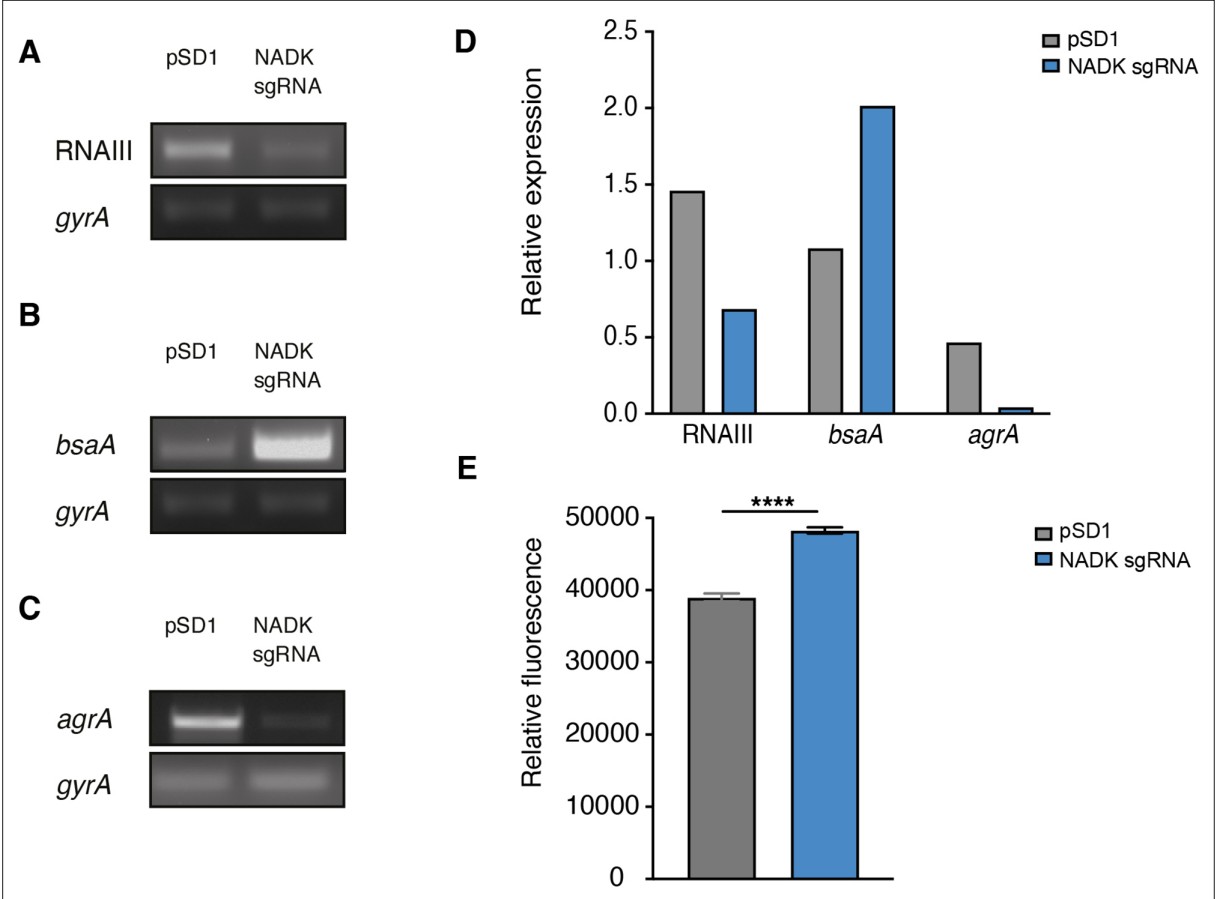

**Figure 5.** Nicotinamide adenine dinucleotide kinase (NADK) is required for AgrA expression and *S. aureus* redox control. The *S. aureus* strain containing the empty vector (pSD1) and the *ppnK* knockdown strain (NADK sgRNA) were grown aerobically in BHI broth at 37°C for 4 hr. (**A–C**) RNA was extracted and subjected to RT-PCR using oligonucleotides specific to *gyrA*, RNAIII, *bsaA*, and *agrA* genes, respectively. (**D**) Relative gene expression was normalized to *gyrA* transcript levels. (**E**) *S. aureus* ROS levels were quantified using the DCFH$_2$-DA assay. Comparison of data was performed using t-test (****p<0.0001). Data are representative of at least three independent experiments.

The online version of this article includes the following source data for figure 5:

**Source data 1.** RNAIII transcript levels (A).

**Source data 2.** BsaA transcript levels (B).

**Source data 3.** AgrA transcript levels (C).

might thus contribute to bacterial survival within macrophages through pathways other than ROS detoxification. We showed that inhibition of *ppnK* expression led to growth defect upon envelope stress, a condition that is encountered in the host, particularly in macrophages where bacteria face antimicrobial peptides and hydrolytic enzymes such as lysozyme (**Rosenberger et al., 2004**; **Cohn and Wiener, 1963**; **Callewaert and Michiels, 2010**). *S. aureus* resists lysozyme through peptidoglycan modification by the *O*-acetyltransferase OatA (**Bera et al., 2005**; **Bera et al., 2006**). Regulation by the glycopeptide resistance-associated two-component system GraRS and D-alanylation of teichoic acid by the *dlt* operon gene products also contribute to *S. aureus* resistance to lysozyme, as well as CAMP (**Herbert et al., 2007**). OatA, DltA, and GraRS were less abundant when NADK was knocked down, suggesting that NADK could help *S. aureus* resist antibacterial envelope-targeting compounds produced by macrophages and/or secreted by the host. Strikingly, our proteomic analysis also highlighted the contribution of NADK to production of virulence factors. Among them, PSM-alpha peptides have previously been shown to lyse leucocytes, elicit phagosomal escape and play a key role in pathogenesis (**Wang et al., 2007**; **Kobayashi et al., 2011**; **Peschel and Otto, 2013**; **Grosz et al., 2014**; **Kitur et al., 2015**). The pore-forming toxin alpha-hemolysin has also been demonstrated to lyse leukocytes and is required for full virulence in mouse, rabbit, and rat models of infection

(*Patel et al., 1987*; *O'Callaghan et al., 1997*; *McElroy et al., 1999*; *Kitur et al., 2015*; *Seilie and Bubeck Wardenburg, 2017*). Gamma-hemolysin leucocidin is leukotoxic as well and contributes to *S. aureus* bacteremia (*Nilsson et al., 1999*; *Spaan et al., 2014*). Control of these factors by NADK could thus promote bacterial survival in macrophages and zebrafish, and trigger macrophage and neutrophil death upon *S. aureus* infection. Remarkably, we found that inhibition of *ppnK* expression decreases abundance of AgrA, the response regulator of *S. aureus agr* system at the center of quorum-sensing and virulence. AgrA directly enhances the transcription of the PSM-alpha genes and governs RNAIII-dependent expression of multiple virulence enzymes including alpha-hemolysin and leukocidins (*Queck et al., 2008*). Thus, the contribution of NADK to production of virulence factors is presumably mediated through AgrA. NADK regulated expression of RNAIII and BsaA similarly to AgrA, corroborating this hypothesis. Besides lowering abundance of AgrA, NADK knockdown led to decreased levels of *agrA* transcript. Increased ROS levels detected in the knockdown strain could account for this regulation as transcription of the *agr* operon is repressed in oxidative conditions (*Rothfork et al., 2004*; *Baker et al., 2010*; *Sun et al., 2012*). Additionally, oxidation leads to formation of an intramolecular disulfide bond between Cys-199 and Cys-228 in the DNA binding domain of AgrA, thereby introducing a conformational change disrupting its association to promoter region of target genes (*Sun et al., 2012*). A recent study confirmed oxidation sensing of the *agr* system and revealed another mode of AgrA regulation by CoAlation of redox-active Cys199 inhibiting its DNA-binding activity (*Baković et al., 2021*).

In summary, our study shows that NADK supports production of protective enzymes, virulence determinants, and the response regulator AgrA, possibly by controlling the bacterial redox state, ultimately promoting *S. aureus* pathogenic potential. NADK inhibition and quorum quenching are attractive therapeutic approaches. Further investigations are now required to fully understand the contribution of bacterial NADKs during infection processes and develop chemical inhibitors that could be used in combination with current antibiotics to fight multidrug resistant bacteria.

# Materials and methods
## Plasmids, bacterial strains, cell lines, and culture onditions
Bacterial strains and plasmids used in this study are listed in *Table 3*. All plasmids were maintained in *Escherichia coli* TOP10. The restriction deficient *Staphylococcus aureus* strain RN4220 was transformed with plasmids isolated from *E. coli*. The *S. aureus* Xen36 strain was transformed with plasmids isolated from the RN4220 strain as previously described (*Gelin et al., 2020*). *E. coli* was grown in lysogeny broth (LB) medium (Difco) with shaking at 200 rpm or on LB agar plates at 37°C. *S. aureus* was grown in brain heart infusion (BHI) broth (BD) or in tryptic soy broth (TSB) (Difco) with

**Table 3.** Strains and plasmids used in this study.

| Strain/plasmid | Relevant characteristics | Reference |
|---|---|---|
| Strains | | |
| *E. coli* strain | | |
| TOP10 | F⁻*mcrA* Δ(*mrr-hsdRMS-mcrBC*) φ80*lacZ*Δ*M15* Δ*lacX74 recA1 araD139* Δ(*ara-leu*) 7,697 *galU galK* λ⁻ *rpsL*(Str^R) *endA1 nupG* | Invitrogen |
| *S. aureus* strains | | |
| Xen36 | Strain derived from a clinical isolate from a bacteremic patient | PerkinElmer, ATCC49525 |
| Xen36/pSD1 | Xen36 strain carrying plasmid pSD1 | *Gelin et al., 2020* |
| Xen36/NADK sgRNA | Xen36 strain carrying plasmid pSD1 *ppnK* | *Gelin et al., 2020* |
| RN4220 | Restriction-deficient strain derived from NCTC 8325–4 | *Kreiswirth et al., 1983* |
| | | |
| Plasmids | | |
| pSD1 | dCas9 ATc-inducible and sgRNA constitutive expression plasmid | *Zhao et al., 2017* |
| pSD1 *ppnK* | pSD1 plasmid carrying sgRNA targeting *S. aureus ppnK* | *Gelin et al., 2020* |

shaking at 200 rpm or on BHI agar plates at 37°C. When required, culture medium was supplemented with antibiotics (carbenicillin 100 µg/mL for *E. coli*, chloramphenicol 15 µg/mL, or anhydrotetracycline 100 ng/mL for *S. aureus*). RAW 264.7 macrophages (TIB-71, ATCC) were cultured in DMEM (Gibco) containing 10% FCS (Gibco). Macrophages were seeded into 24-well plates at $10^5$ cells per well.

## Bacterial growth measurement

*S. aureus* strains were grown overnight in BHI broth at 37°C and diluted to $OD_{600nm}=0.05$ into BHI broth. For NADK inhibition experiments, culture medium was supplemented with 100 ng/mL anhydrotetracycline (Sigma-Aldrich) and 15 µg/mL chloramphenicol when required. For stress experiments, culture medium was supplemented with hydrogen peroxide (0–500 µM), catalase (10 µg/mL), lysozyme (5 mg/mL), or lysostaphin (0.05 mg/mL). Bacterial suspensions were incubated in 96-well plates with shaking at 200 rpm at 37°C. Growth was monitored with a microplate reader (Glomax Discover, Promega). Experiments were performed at least three times.

## Christie-Atkins-Munch-Peterson test

*S. aureus* hemolytic activity was performed as described previously (*Adhikari et al., 2007*). Briefly, *S. aureus* RN4220 was streaked vertically at the center of a sheep blood agar plate. *S. aureus*/pSD1 and *S. aureus*/NADK sgRNA strains were then streaked perpendicularly to the RN4220 strain. The plate was incubated at 37°C for 18 hr. Blood agar plates were then photographed with a Scan500 imager (Interscience). The assay was repeated three times.

## RNA extraction and RT-PCR

Bacteria grown for 4 hr were collected by centrifugation 20 min at 15,000 × *g* at 4°C. Bacterial lysis was performed using Precellys lysing kit (P000914-LYSK0-A) and the Precellys program 4 (6.500 rpm for 30 s) twice at 4°C. RNA extraction was performed in 1 mL of TRIzol (Invitrogen) according to the manufacturer's recommendations. RT-PCR was carried out with Superscript one-step RT kit (Invitrogen). Primers used are indicated in *Table 4*. RT-PCR assays were repeated at least three times.

## Immunoblotting

Bacteria grown for 4 hr were collected by centrifugation 10 min at 10,000 × *g* at 4°C and resuspended in 500 µL of PBS. Bacteria were lysed using the Precellys lysing kit (P000914-LYSK0-A) and the Precellys program 4 (6500 rpm for 30 s) twice at 4°C. Lysates were centrifuged for 15 min at 10,000 × *g* at 4°C and supernatants were collected for protein quantification using the Quick start Bradford protein assay kit 2 (Biorad). Samples were mixed with Laemmli buffer (Biorad) and 10% β-mercaptoethanol and denatured for 5 min at 95°C. Samples were separated onto 4–20% Miniprotean TGX stain-free precast gel (Biorad) in TGS buffer and transferred on polyvinylidene fluoride membranes. Membranes

**Table 4.** Primers used in this study.

| Primer | Sequence (5'–3') |
|---|---|
| RT-ppnK-F | GTGACTCCAAGTCTAATGCC |
| RT-ppnK-R | ATTTTTCAACTTCATGAGGTAACC |
| RT-gyrA-F | GTGTTATCGTTGCTCGTG |
| RT-gyrA-R | CGGTGTCATACCTTGTTC |
| RT-RNAIII-F | AGTTTCCTTGGACTCAGTGCT |
| RT-RNAIII-R | AGGGGCTCACGACCATACTT |
| RT-bsaA-F | GCGAAGAAGCAGCTCAAAAC |
| RT-bsaA-R | CCTTCGCGATCCACTAAAAA |
| RT-agrA-F | GCCCTCGCAACTGATAATCC |
| RT-agrA-R | CACCGATGCATAGCAGTGTTC |
| RT-rluA-F | CCGCGAGAAATACCGAGTGT |
| RT-rluA-R | TCTCCCACAATTGGATGCCC |
| RT-operon-F1 | TGACTTGCTTAAAAAGCACACTG |
| RT-operon-R1 | ACGAGCATTTGTCCTACTTCAGA |
| RT-operon-F2 | AACCGTTGAAGAAACATTCGACA |
| RT-operon-R2 | GACGCTTGTTCACCAACTTCA |
| RT-operon-F3 | CATCGTTTGGAAAGAGCGGC |
| RT-operon-R3 | GGCATTAGACTTGGAGTCACCT |
| RT-operon-F4 | ACGTGTGCACGATTCTTTCAT |
| RT-operon-R4 | ATGGCGCTCACTGTCTTCT |
| RT-operon-F5 | AGTTCATTTGCATACGGGACG |
| RT-operon-R5 | ACGCTCTTTTTCATCTGTGTTCA |
| RT-operon-F6 | TAACTTGTGCGATGACGGTGG |
| RT-operon-R6 | TTCCAATCACAATCCCCATCAA |
| RT-operon-F7 | ACGTTGATGAATTGAAGCAAGAG |
| RT-operon-R7 | ACTTTAGCGACACCAAAAGCA |
| RT-operon-F8 | TCAAGTGGCGTTACAGGTGA |
| RT-operon-R8 | TTCAAATACCGCCAACGCAT |

were incubated overnight at 4°C with primary antibodies diluted in 5% blotto (R114 rabbit anti-EF-Tu polyclonal antibodies, 1:5000; R250 rabbit polyclonal anti-*S. aureus* NADK antibodies, 1:1000; anti-staphylococcal α-toxin (α-hemolysin) rabbit antiserum (Sigma-Aldrich S7531) 1:1000). Membranes were then incubated with 1:2500 antirabbit horseradish peroxidase-conjugated antibodies (Abcam). Blots were revealed using the ECL kit (Pierce).

## RAW 264.7 macrophage infection

RAW 264.7 macrophages were seeded into 24-well plates at 100,000 cells per well. *S. aureus* Xen36/pSD1 and *S. aureus* Xen36/NADK sgRNA strains were grown to an $OD_{600nm}$ of 0.6, washed three times and diluted in DMEM to obtain a multiplicity of infection (MOI) of 10. Infected macrophages were centrifugated at $500 \times g$ for 5 min and incubated for 15 min at 37°C and 5% $CO_2$ to synchronize phagocytosis. The medium was replaced with DMEM containing gentamicin (20 µg/mL) for 30 min to kill extracellular bacteria. Cells were washed three times in DPBS and incubated in DMEM supplemented with 100 ng/mL anhydrotetracycline and 10 mM N-acetyl cysteine (NAC, Sigma-Aldrich) when indicated. Cells were incubated for 6 hr at 37°C and 5% $CO_2$. At each time point, cells were lysed in 0.2% Triton X-100 for 10 min at 37°C. The number of bacteria released from the cells was determined by plating serial dilutions of the lysates on BHI agar plates that were incubated at 37°C for colony-forming units (CFU) enumeration.

## Immunofluorescence assay

RAW 264.7 macrophages were cultured on glass coverslips in 24-well plates. At 6 hr post-infection, cells were fixed in paraformaldehyde (PFA) 4% for 15 min, permeabilized for 4 min with 0.1% Triton X-100 in 1% bovine serum albumin (BSA)-PBS and blocked in 1% BSA-PBS. Fixed cells were incubated for 30 min with rabbit anti-*S. aureus* (Abcam ab20920) and rat anti-LAMP1 1D4B (Abcam ab25245) primary antibodies. Cells were then incubated for 30 min with goat anti-rabbit Cy5 (Immunoresearch 111 175 144) or goat anti-rat FITC (Invitrogen A11006) secondary antibodies and DAPI (10 µg/mL). Slides were mounted with Fluoromount G (Invitrogen). Samples were observed with a Zeiss Axiovert 200 M epifluorescence microscope equipped with a Plan-apochromat objective (100 X/1.4 Oil Ph3; Carl Zeiss, Inc). Images were acquired with a CCD camera Coolsnap, processed with Metamorph software v.6 (Molecular Devices), and analyzed with ImageJ software.

## Cytotoxicity assay

Overnight cultures of *S. aureus*/pSD1 and *S. aureus*/NADK sgRNA were diluted to an $OD_{600nm}$ 0.05 in BHI supplemented with ATc (100 ng/mL) and chloramphenicol (7 µg/mL), spread in a 96 wells plate and incubated for 6 hr at 37°C with shaking at 200 rpm. After 5hr 30min of growth, lysozyme (5 mg/mL) and lysostaphin (0.05 mg/mL) were added. At t=6 hr, 100 µL of the CellTox Green Reagent (1:500, G8741, Promega) were added to bacteria. After incubation at RT for 15 min, both $OD_{600nm}$ and fluorescent signal (excitation at 475 nm, emission at 500–550 nm) were measured with a microplate reader (Glomax Discover, Promega).

For the cytotoxicity assay on infected macrophages, culture medium was replaced by either DPBS or by 0.2% Triton X-100 6 hr after the infection. After 10 min at 37°C, cells were washed with DPBS, and then incubated with 100 µL of CellTox Green Reagent for 15 min at RT. Fluorescent signal (excitation at 475 nm, emission at 500–550 nm) was measured with a microplate reader (Glomax Discover, Promega).

## ROS levels quantification

Overnight cultures of *S. aureus*/pSD1 and *S. aureus*/NADK sgRNA were diluted to an $OD_{600nm}$ 0.05 in BHI supplemented with ATc (100 ng/mL) and chloramphenicol (7 µg/mL), spread in a 24-wells plate and incubated at 37°C with shaking at 200 rpm for 4 hr. After 4 hr, the $OD_{600nm}$ was measured and adjusted to 1, then bacteria centrifuged for 3 min at 13.000 g. $DCFH_2$-DA (5 mM) was mixed with NaOH (0.1 M) to obtain DCF. Bacterial pellets were resuspended in 100 uL of DCF and incubated for 40 min at RT protected from light in a 96-wells black plate. Then fluorescence signal (excitation 475 nm, emission 500–550 nm) was measured with a microplate reader (Glomax Discover, Promega).

## Zebrafish care and maintenance

Homozygous Tg(*mfap4::mCherryF*) (ump6Tg) (*Phan et al., 2018*) Tg(*mpx::GFP*)[i114] (*Renshaw et al., 2006*) double transgenic fishes were raised in our facility. Eggs were obtained by natural spawning, bleached according to standard protocols, and then kept in Petri dishes containing Volvic spring water and, from 24 hpf onwards, 0.003% 1-phenyl-2-thiourea (PTU) (Sigma-Aldrich) was added to prevent pigmentation. Embryos were reared at 28 or 24°C according to the desired speed of development; infected larvae were always kept at 28°C. All timings in the text refer to the developmental stage at the reference temperature of 28°C. Larvae were anesthetized with 200 µg/mL of buffered tricaine methane sulfonate (MS-222, Sigma-Aldrich) during the injection procedure as well as during in vivo imaging and processing for bacterial burden evaluation.

## Zebrafish infection

The volume of injected bacterial suspension was deduced from the diameter of the drop obtained after mock microinjection, as previously described (*Levraud et al., 2008*). Bacteria were diluted from overnight cultures and allowed to reach exponential growth. Then, they were recovered by centrifugation, washed, and resuspended at the desired concentration in PBS. 55–60 hr post-fertilization, anesthetized zebrafish larvae were microinjected intravenously (IV) with 1 nL of bacterial suspension at the desired dose as described (*Colucci-Guyon et al., 2011*; *Mostowy et al., 2013*). Infected larvae were transferred into individual wells containing 1 mL of Volvic water and 0.003% PTU in 24-well culture plates, incubated at 28°C, and regularly observed under a stereomicroscope.

## Evaluation of the bacterial burden in infected zebrafish larvae

Infected zebrafish larvae were collected at 0, 24, 48, and 72 hpi and lysed for the evaluation of the bacterial burden as previously described (*Boucontet et al., 2018*; *Mostowy et al., 2013*). Each larva was placed in an individual 1.5 mL Eppendorf tube and anesthetized with tricaine (200 µg/mL), washed with 1 mL of sterile water, and placed in 150 µL of sterile water. Larvae were then homogenized using a pestle motor mixer (Argos). Each sample was transferred to an individual well of a 96-well plate and 10 X serial dilutions were performed. For CFU enumeration and to assess plasmid stability throughout the infection kinetics, serial dilutions of lysates were plated on BHI agar plates with and without chloramphenicol (15 µg/mL), that were incubated overnight at 37°C.

## Zebrafish live imaging, image processing, and analysis

Quantification of total macrophages and neutrophils on living transgenic reporter larvae was performed upon infection as we previously described (*Mostowy et al., 2013*). Briefly, bright field, GFP, and RFP images of whole living anesthetized larvae were taken using a Leica Macrofluo Z16 APOA (zoom 16:1) macroscope equipped with a Leica PlanApo 2.0 X lens, and a Photometrics CoolSNAP *HQ2* camera. Images were captured using the Metavue software version 7.5.6.0 (MDS Analytical Technologies). After capture of images, larvae were washed and transferred to a new 24-well plate filled with 1 mL of fresh water per well and incubated at 28 °C.

## Whole-mount immunohistochemistry and confocal imaging

Whole larvae were fixed in 4% formaldehyde/PBS overnight at 4°C. After washes in PBSDT (PBS, 1% DMSO, and 0.1% Tween), fixed larvae were treated with acetone at –20° for 20 min, rinsed with 1×HBSS/0.1% Tween, permeabilized with a collagenase solution for up to 1 hr (*Svoboda et al., 2001*), washed in PBSDT, blocked with 10% Western Blocking Reagent (Merck; 11921673001)/PBSDT, and then incubated for 24–48 hr at 4°C with the primary antibodies. Chicken anti-GFP (1:1000, ab13970, Abcam), and anti-*Staphylococcus aureus* (1:500, ab20920, Abcam) antibodies were used. Following washes, samples were blocked in 10% Western blocking solution/PBSDT before overnight incubation with secondary antibodies at 4°C with agitation. The secondary antibodies were Alexa Fluor 488-conjugated goat anti-chicken (1:1000, A11039, Invitrogen) and Cy5-conjugated goat anti-rabbit (1:200, 111-175-144, Jackson Immunoresearch). Larvae were progressively embedded in glycerol (for conservation and microscopic observation) and mounted in lateral position for imaging. Confocal fluorescent imaging of fixed injected larvae was performed as previously described (*Colucci-Guyon et al., 2020*). Briefly, injected larvae were positioned in lateral position in 35 mm glass-bottom µ-Dishes (81158, Ibidi). Larvae were immobilized in the dish using a drop of 1% low-melting-point agarose

(V2111, Promega) solution in 80% glycerol. A Leica SP8 confocal microscope equipped with two PMT and one Hybrid detector, a 20 X multi-IMM (water-oil-glycerol) objective (HC PL APO CS2 20 X/0.75), a X–Y motorized stage and the LAS-X software was used to image injected larvae. All samples were acquired using the same settings.

## Total protein extraction

For in vitro samples, overnight cultures of *S. aureus*/pSD1 and *S. aureus*/NADK sgRNA were diluted to an $OD_{600nm}$ 0.05 in BHI with 200 µM $H_2O_2$ when indicated, spread in a 24-well plate, and incubated at 37°C with shaking at 200 rpm. After 6 hr, bacteria were centrifuged for 15 min at 10,000 *g* at 4°C, and the bacterial pellets were stored at –80°C. For in vivo samples, zebrafish larvae were infected with *S. aureus*/pSD1 or *S. aureus*/NADK sgRNA strains (see above). At 12 hpi, 5 larvae of each condition were isolated, anesthetized with tricaine, washed with 1 mL of sterile water, dried, placed on dry ice, and stored at –80°C. All samples were then resuspended in 1 mL of Tris 100 mM pH 7/urea 8 M. Proteins were extracted using the Precellys kit SK38 (P000915-LYSK0-A0, Bertin), in a Cryolys homogenizer (Bertin) by 6 cycles of agitation at 6500 rpm for 15 s and 30 s rest.

## Protein digestion and cleanup of peptides

Proteins were reduced using 5 mM TCEP for 30 min at room temperature. Alkylation of the reduced disulfide bridges was performed using 10 mM iodoacetamide for 30 min at room temperature in the dark. Proteins were then digested in two steps, first with 1 µg r-LysC Mass Spec Grade (Promega) for 4 hr at 30°C and then samples were diluted below 2 M urea with 100 mM Tris HCl pH 8.5 and 1 µg Sequencing Grade Modified Trypsin was added for the second digestion overnight at 37°C. Proteolysis was stopped by adding formic acid (FA) at a final concentration of 5%. The peptide samples were cleaned using $C_{18}$ cartridges (Agilent Technologies) in the automated AssayMAP Bravo Platform. The protocol for peptide cleanup included in the platform was used as a scaffold using the following settings. Cartridges were primed with 100 µL of 100% acetonitrile (ACN) at 300 µL/min. Equilibration was done with 50 µL of 0.1% FA at 10 µL/min. The sample was loaded at 5 µL/min. Cup wash (25 µL) and the internal cartridge wash (50 µL at 10 µL/min) were performed with 0.1% FA. Peptides were eluted with 50 µL of 50% ACN/0.1% FA at 5 µL/min. Samples were dried and stored at −80°C until further use. Peptides were resuspended in 2% acetonitrile (ACN)/0.1% FA prior to LC-MS injection.

## Liquid chromatography-mass spectrometry

LC-MS/MS analysis was performed on a Orbitrap Q Exactive Plus Mass Spectrometer (Thermo Fisher Scientific) coupled with a Proxeon EASY-nLC 1200 (Thermo Fisher Scientific). One µg of peptides was injected onto a home-made 30 cm $C_{18}$ column (1.9 µm particles, 100 Å pore size, ReproSil-Pur Basic C18, Dr. Maisch GmbH, Ammerbuch-Entringen, Germany). Column equilibration and peptide loading were done at 900 bars in buffer A (0.1% FA). Peptides were separated with a multi-step gradient from 2 to 7% buffer B (80% ACN, 0.1% FA) in 5 min, 7 to 23% buffer B in 70 min, 23 to 45% buffer B in 30 min at a flow rate of 250 nL/min. Column temperature was set to 60°C. MS data were acquired using Xcalibur software using a data-dependent method. MS scans were acquired at a resolution of 70,000 and MS/MS scans (fixed first mass 100 m/z) at a resolution of 17,500. The AGC target and maximum injection time for the survey scans and the MS/MS scans were set to $3E^6$, 20ms, and $1E^6$, 60ms, respectively. An automatic selection of the 10 most intense precursor ions was activated (Top 10) with a 45 s dynamic exclusion. The isolation window was set to 1.6 m/z and normalized collision energy fixed to 28 for HCD fragmentation. We used an underfill ratio of 1.0% corresponding to an intensity threshold of $1.7E^5$. Unassigned precursor ion charge states as well as 1, 7, 8, and >8 charged states were rejected and peptide match was disabled.

## Data analysis

Acquired Raw files were analyzed using the MaxQuant software version 1.6.6.0 (*Cox et al., 2011*) using the Andromeda search engine (*Cox and Mann, 2008*; *Tyanova et al., 2016*). Sample were grouped by type of experiment. The MS/MS spectra were searched against the *S. aureus* Uniprot reference proteome (2889 entries).

All searches were performed with oxidation of methionine and protein N-terminal acetylation as variable modifications and cysteine carbamidomethylation as fixed modification. Trypsin was selected

as protease allowing for up to two missed cleavages. The minimum peptide length was set to five amino acids and the peptide mass was limited to a maximum of 8000 Da. The false discovery rate (FDR) for peptide and protein identification was set to 0.01. The main search peptide tolerance was set to 4.5 ppm and to 20 ppm for the MS/MS match tolerance. Second peptides were enabled to identify co-fragmentation events. One unique peptide to the protein group was required for the protein identification. A false discovery rate cut-off of 1% was applied at the peptide and protein levels. The mass spectrometry proteomics data have been deposited to the ProteomeXchange Consortium via the PRIDE partner repository with the dataset identifier PXD032968.

The statistical analysis of the proteomics data was performed as follows: Five biological replicates were acquired per condition. To highlight significantly differentially abundant proteins between two conditions, differential analyses were conducted through the following data analysis pipeline: (1) deleting the reverse and potential contaminant proteins; (2) keeping only proteins with at least three quantified values in one of the two compared conditions to limit misidentifications and ensure a minimum of replicability; (3) log2-transformation of the remaining intensities of proteins; (4) normalizing the intensities by median centering within conditions thanks to the *normalizeD* function of the R package DAPAR (*Wieczorek et al., 2017*), (5) putting aside proteins without any value in one of both compared conditions: as they are quantitatively present in a condition and absent in another, they are considered as differentially abundant proteins, and (6) performing statistical differential analysis on them by requiring a minimum fold-change of 2.5 between conditions and by using a LIMMA t-test (*Smyth, 2005*; *Ritchie et al., 2015*) combined with an adaptive Benjamini-Hochberg correction of the p values thanks to the *adjust.p* function of the R package cp4p (*Giai Gianetto et al., 2016*). The robust method of Pounds and Cheng was used to estimate the proportion of true null hypotheses among the set of statistical tests (*Pounds and Cheng, 2006*). The proteins associated with an adjusted p-value inferior to an FDR level of 1% have been considered as significantly differentially abundant proteins. Finally, the proteins of interest are therefore the proteins that emerge from this statistical analysis supplemented by those being quantitatively absent from one condition and present in another.

## Statistical analysis

Results are expressed as means ± SEM of at least three replicates. Statistical analysis was performed using GraphPad Prism (GraphPad Prism 9.1.2. Software). One-way, two-ways ANOVA or t-test were used to compare data. Differences between groups were considered significant when the p-value was lower than 0.05. Survival data were plotted using the Kaplan-Meier estimator and log-rank (Mantel-Cox) tests were performed to assess differences between groups.

## Acknowledgements

Authors are deeply indebted to Liliane Assairi, Sylvie Pochet and Gilles Labesse who led us into the world of NAD kinases. We are grateful to past and present members of the NADK consortium, all members of the *Yersinia* research unit and the French national reference center for plague and other yersiniosis, as well as Frédéric Barras and Daniel Ladant for helpful discussions and insightful comments. We thank Changlong Zhao for providing pSD1. We also wish to thank the members of the fish facility team Yohann Rolin and Noël Aimar for their excellent care of the fish. This work was funded by the Agence Nationale de la Recherche (grant ANR-17-CE18-0011-02) and supported by Institut Pasteur, Université Paris Cité and CNRS. The *Yersinia* Research Unit is member of the Laboratory of Excellence Integrative Biology of Emerging Infectious Diseases (ANR-10-LBX-62-IBEID). We are grateful for support for equipment from the French government Programme d'investissements d'avenir France BioImaging (ANR-10-INSB-04–01).

## Additional information

### Funding

| Funder | Grant reference number | Author |
|---|---|---|
| Agence Nationale de la Recherche | ANR-17-CE18-0011-02 | Clarisse Leseigneur<br>Olivier Dussurget |
| Agence Nationale de la Recherche | ANR-10-LBX-62-IBEID | Clarisse Leseigneur<br>Javier Pizarro-Cerda<br>Olivier Dussurget |
| Agence Nationale de la Recherche | ANR-10-INSB-04-01 | Laurent Boucontet<br>Emma Colucci-Guyon |
| Institut Pasteur | | Clarisse Leseigneur<br>Olivier Dussurget<br>Javier Pizarro-Cerda<br>Laurent Boucontet<br>Emma Colucci-Guyon<br>Magalie Duchateau<br>Mariette Matondo |
| Agence Nationale de la Recherche | | Clarisse Leseigneur<br>Olivier Dussurget<br>Javier Pizarro-Cerda<br>Laurent Boucontet<br>Emma Colucci-Guyon |
| Centre National de la Recherche Scientifique | | Clarisse Leseigneur<br>Olivier Dussurget<br>Javier Pizarro-Cerda<br>Laurent Boucontet<br>Emma Colucci-Guyon<br>Magalie Duchateau<br>Mariette Matondo |

The funders had no role in study design, data collection and interpretation, or the decision to submit the work for publication.

### Author contributions

Clarisse Leseigneur, Conceptualization, Formal analysis, Investigation, Methodology, Resources, Validation, Writing - original draft, Writing - review and editing; Laurent Boucontet, Mariette Matondo, Emma Colucci-Guyon, Formal analysis, Investigation, Methodology, Resources, Supervision, Validation, Writing - review and editing; Magalie Duchateau, Formal analysis, Investigation, Methodology, Resources, Validation, Writing - review and editing; Javier Pizarro-Cerda, Formal analysis, Resources, Writing - review and editing; Olivier Dussurget, Conceptualization, Data curation, Formal analysis, Funding acquisition, Investigation, Methodology, Project administration, Resources, Supervision, Validation, Writing - original draft, Writing - review and editing

### Author ORCIDs

Clarisse Leseigneur ![ORCID] http://orcid.org/0000-0001-8666-9095
Laurent Boucontet ![ORCID] http://orcid.org/0000-0001-6047-7028
Magalie Duchateau ![ORCID] http://orcid.org/0000-0001-5475-3065
Javier Pizarro-Cerda ![ORCID] http://orcid.org/0000-0002-4343-0508
Mariette Matondo ![ORCID] http://orcid.org/0000-0003-3958-7710
Olivier Dussurget ![ORCID] http://orcid.org/0000-0001-9924-0837

### Ethics

Animal experiments were performed according to European Union guidelines for handling of laboratory animals (https://ec.europa.eu/environment/chemicals/lab_animals/index_en.htm). All experiments performed on larvae older than 5 days post fertilization were approved by the Institut Pasteur Animal Care and Use Committee and the French Ministry of Education, Research and Innovation, and registered under the reference APAFIS#31827.

Decision letter and Author response
Decision letter https://doi.org/10.7554/eLife.79941.sa1
Author response https://doi.org/10.7554/eLife.79941.sa2

## Additional files

### Supplementary files

• Supplementary file 1. Proteins more abundant in the *S. aureus* pSD1 strain than in the NADK sgRNA strain after growth in BHI broth.

• Supplementary file 2. Relative abundance of a subset of protective proteins differentially expressed by *S. aureus* pSD1 and NADK sgRNA strains exposed to hydrogen peroxide.

• Supplementary file 3. Proteins more abundant in the *S. aureus* pSD1 strain than in the NADK sgRNA strain after growth in BHI broth supplemented with hydrogen peroxide.

• Supplementary file 4. Relative abundance of a subset of protective proteins differentially expressed by *S. aureus* pSD1 and NADK sgRNA strains upon zebrafish infection.

• Supplementary file 5. Proteins more abundant in the *S. aureus* pSD1 strain than in the NADK sgRNA strain upon zebrafish infection.

• Supplementary file 6. Relative abundance of a subset of virulence-related proteins differentially expressed by *S. aureus* pSD1 and NADK sgRNA strains exposed to hydrogen peroxide.

• MDAR checklist

### Data availability

Proteome data have been submitted to ProteomeXchange via the PRIDE repository. A proteome data access code file has been provided.

The following dataset was generated:

| Author(s) | Year | Dataset title | Dataset URL | Database and Identifier |
|---|---|---|---|---|
| Dussurget O | 2022 | NAD kinase promotes *Staphylococcus aureus* pathogenesis by supporting production of virulence factors and protective enzymes | https://www.ebi.ac.uk/pride/archive/projects/PXD032968 | PRIDE, PXD032968 |

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
