## [Editor Report]

This work demonstrates that nicotinamide adenine dinucleotide phosphate is essential for the virulence of *Staphylococcus aureus*. The enzyme that catalyses the final step in the synthesis of this cofactor promotes virulence by protecting bacteria from host antimicrobial defences and is also involved in quorum sensing mechanisms. These data shed new light on the pathogenesis of *S. aureus*.

---

## [Decision Letter]

**Decision letter after peer review:**

[Editors’ note: the authors resubmitted a revised version of the paper for consideration. What follows is the authors’ response to the first round of review.]

Thank you for submitting the paper "NAD kinase controls antibiotic susceptibility and pathogenic potential in *Staphylococcus aureus*" for consideration by *eLife*. Your article has been reviewed by 2 peer reviewers, and the evaluation has been overseen by a Reviewing Editor and a Senior Editor. The reviewers have opted to remain anonymous.

We are sorry to say that, after consultation with the reviewers, we have decided that this work will not be considered further for publication by *eLife*.

Your manuscript raised a very interesting model about how pink knock-down sensitizes bacteria to several stresses or more specifically ROS and is constituted of an incremental complexity of experiments from in vitro to in vivo. Unfortunately, it falls short of providing mechanistic insights on to the observations and alternatives are not considered. We recommend using the reviews provided here to consolidate the findings before submitting to another journal.

*Reviewer #1:*

The authors set out to examine the role of the enzyme NADK in enabling *S. aureus* to cause infection and survive antibiotic exposure.

The main strength is that the authors managed to reduce the expression of NADK in *S. aureus* and go on to produce some interesting findings. The main weaknesses are that the work lacks depth and a number of important controls. There is limited investigation of the mechanisms that underpin the phenotype, resulting in a very superficial analysis of each phenotype. Importantly, alternative explanations for their observations are not investigated. The magnitude of differences in antibiotic susceptibility seems very small and of questionable physiological or clinical relevance.

The overall aim was partly achieved in that the authors showed that NADK influences pathogenesis but the data on antibiotic susceptibility is very weak. Conclusions on the role of NADK in antibiotic susceptibility and ROS susceptibility are compromised by the lack of control and approaches used.

It's not clear why the authors used the Xen36 strain or whether the findings could be applied to other strains?

Figure supplement 1 shows the relQ operon for NCTC8325. Is this identical to the operon in Xen36? If so, why show for NCTC3525?

Line 123: It would be useful to estimate the degree to which ppnK expression and NADK production were knocked down.

Lines 122-123 fail to mention the growth defect of the KD strain, which is important and could impact many other phenotypes. Given the relationship of NADK to oxidative stress, it would be useful to see if this growth defect is abolished under anaerobic conditions. This will determine whether growth is due to increase susceptibility to oxidative stress or e.g. decreased metabolic activity.

Knockdown of NADK production sensitises *S. aureus* to H2O2, but it's not clear why. It would be useful to show that catalase production is unaffected in the KD strain.

Lines 153-154: For the triclosan susceptibility experiments, the effect shown in panel 3A is not obviously different between the KD strain and control. It doesn't appear that any statistical tests were done here to show differences (the only comparisons appear to be to No Treatment, not between strains). For panel B, it's unclear why catalase promotes growth of the KD strain. Is this strain catalase deficient?

The EM analysis in figure supplement 2D isn't particularly convincing since most of the cells look fine. It could be that the images represent debris from the culture. A better, more quantitative test of membrane integrity would be to use a stain such as propidium iodide. It would also be useful to examine bacterial susceptibility to agents that target the membrane such as gramicidin, nisin or daptomycin. I note the subsequent experiment with polymyxin B, but this is typically used to combat Gram-negative bacteria as it has a high affinity for LPS and it's not clear what the concentrations used in this work do to the bacterial membrane, especially since there's only ~20% growth inhibition.

Line 189 refers to bacterial survival during antibiotic stress, but the assay measures growth. These are two very different things. Survival is usually measured by time-course killing assays and growth by determination of the minimum inhibitory concentration.

For the antibiotic assays it is not clear to me how the antibiotics were chosen. Polymyxin B is not used to treat *S. aureus* infections, nor is kanamycin or levofloxacin used routinely. It would be informative to include an ant-staphylococcal penicillin such as oxacillin. I'm also unsure of the relevance of this assay. It seems like there are very small differences in susceptibility between KD and control strain and the drug concentrations used are chosen specifically to focus on these small differences. If the author wants to investigate the role of ROS, they should repeat experiments under anaerobic conditions.

Line 194: All Gram-positive bacteria are resistant to polymyxins since they lack LPS. Activity against Gram-positives is thought to be due to the surfactant properties of the antibiotic but this is not well established. A much better choice here would be daptomycin since it is used against *S. aureus* and targets the membrane.

When measuring survival of *S. aureus* in phagocytes, it is important to consider whether there are differences in production of cytolytic toxins, which can disrupt immune cell function. Quantitative measurements of hemolytic activity or π staining of immune cells would suffice. This is especially important since the author observed escape from phagosomes, which is likely toxin mediated.

The zebrafish data could be a consequence of different growth rates of the KD and control strain or differences in toxin production. In fact, toxin production seems more likely since relatively high CFU were found in fish, but the fish survived. Furthermore, there seems to be a depletion of phagocytic cells. Therefore, it's important to determine whether there are differences in toxin production between strains. If the authors wish to confirm the importance of oxidative immune defences, they should use a fish line defective for the oxidative burst.

Figure 6A, B doesn't appear to have been analysed using any statistical tests.

Line 317. ROS levels were not measured so this conclusion is unsupported. There are several other antibacterial factors in macrophages, including antimicrobial peptides that target the bacterial membrane.

Line 345: If an inhibitor of NADK is available this should be used to confirm key findings.

Line 392. What time point was used for antibiotic-mediated growth inhibition experiments and why?

*Reviewer #2:*

This work describes the role of Staphylococcal NADK in bacterial pathogenesis. It uses CRISPR-based interference to knock-down expression of the ppnK gene that encodes for NADK in *Staphylococcus aureus*. Comparing the behaviour of wild-type and ppnK knock-down *S. aureus*, the authors show that NADK protects the bacteria against oxidative stresses directly induced by 1) peroxide, 2) inhibition of fatty acid elongation, 3) exposure to various classes of antibiotics, and 4) uptake by macrophages. In addition, they show that NADK protects against some non-oxidative stresses like osmotic shock and exposure to other antibiotics. Lastly, they show that NADK knockdown impairs *S. aureus* survival in macrophages and impairs pathogenicity in a zebrafish infection model.

The observations in the paper are well described. However, more technical detail is required to unequivocally support the major conclusion.

Specifically, the authors start with showing that ppnK is part of an operon. Therefore, targeting ppnK via CRISPR-based interference could also affect genes downstream in this operon as described by Zhao et al. who devised the system (described in Applied and Environmental Microbiol 2017). To unequivocally show that NADK mediates the observed phenotypes, the authors should complement their knockdown strain with exogenous ppnK expression, or measure and complement NADP(H) levels.

The CRISPR-based interference that is used depends on a vector that allows for constitutive expression of sgRNA and inducible expression of dCas9 to mediate the interference. Zhao et al. described that ATc induction leads to a 100-fold knockdown of a target gene. The knockdown is reversible and non-induced leaky expression will still lead to 3-fold reduction in expression of the target gene. The authors should comment on what this means for the NADK expression during the macrophage and zebrafish infection experiments that run over longer periods. Perhaps the authors know whether initial absence of NADK during infection is sufficient to evoke the better survival of fish at 72h? Or whether the non-induced strain is also less pathogenic?

1. The way the introduction is written, it is not directly clear whether the paper would focus on bacterial or human NADK. Stating this earlier could make the paper easier to read.

2. Looking at the described experiments and the points raised in the general discussion, the regulation of NADKs by calcium and calmodulin seem redundant. Perhaps the authors can clarify why they highlight this in the introduction?

3. Readers would benefit from a broader perspective in the discussion about known ways in which *S. aureus* deals with ROS during infection. For example, production of catalase, staphyloxanthin, and the MPO inhibitor SPIN.

4. Line 111: to my opinion, this finding does not "confirm importance", but suggests it.

---

## [Author Response]

Your manuscript raised a very interesting model about how pink knock-down sensitizes bacteria to several stresses or more specifically ROS and is constituted of an incremental complexity of experiments from in vitro to in vivo. Unfortunately, it falls short of providing mechanistic insights on to the observations and alternatives are not considered. We recommend using the reviews provided here to consolidate the findings before submitting to another journal.Reviewer #1:The authors set out to examine the role of the enzyme NADK in enabling *S. aureus* to cause infection and survive antibiotic exposure.The main strength is that the authors managed to reduce the expression of NADK in *S. aureus* and go on to produce some interesting findings. The main weaknesses are that the work lacks depth and a number of important controls. There is limited investigation of the mechanisms that underpin the phenotype, resulting in a very superficial analysis of each phenotype. Importantly, alternative explanations for their observations are not investigated. The magnitude of differences in antibiotic susceptibility seems very small and of questionable physiological or clinical relevance.The overall aim was partly achieved in that the authors showed that NADK influences pathogenesis but the data on antibiotic susceptibility is very weak. Conclusions on the role of NADK in antibiotic susceptibility and ROS susceptibility are compromised by the lack of control and approaches used.

We thank the reviewer for highlighting the production of interesting findings by reducing NADK expression in *S. aureus*. We have worked to address suggestions on depth, controls, mechanisms, alternatives, and antibiotic susceptibility.

It's not clear why the authors used the Xen36 strain or whether the findings could be applied to other strains?

The Xen36 strain has been used for several reasons. Xen36 is a relevant strain to study pathogenicity as it is derived from a clinical strain isolated from a bacteremic patient (ATCC 49525). This is indicated in Table 3 (page 25). As a matter of fact, we and many others have been using this strain to investigate *S. aureus*-host interactions for more than a decade (Koskinen et al., 2007; Munro et al., 2010; Pribaz et al., 2012; Peetermans et al., 2014; Bostian et al., 2017; Trikha et al., 2020 etc.). Additionally, Xen36 is commercially available and can be readily obtained from PerkinElmer, as indicated in Table 3 (page 25). Most importantly, we previously showed that we could modify genetically this strain to knockdown NADK expression (Gelin et al., 2020). This is mentioned lines 94-96.

To assess if our findings could be applied to other strains, we compared growth of the S. aureus USA300 strain containing the empty vector to that of the NADK knockdown strain upon exposure to hydrogen peroxide. USA300 is a globally distributed, methicillin-resistant, hypervirulent strain of community- and healthcare-associated *S. aureus* (Planet, 2017). As expected, growth of USA300 in BHI was affected by NADK knockdown (Author response image 1). When bacteria were exposed to a 200 μM concentration of H2O2, the USA300/pSD1 strain was not affected. In contrast, the USA300/NADK sgRNA strain showed a significant growth defect ( (Author response image 1) ). These results indicate that NADK protects USA300 from hydrogen peroxide toxicity, as demonstrated in Xen36 (Figure 3, page 14). Therefore, our findings could be applied to both methicillin-sensitive and MRSA strains.

**Author response image 1. sa2fig1:** NADK protects *S. aureus* USA300 from hydrogen peroxide toxicity. Growth of the USA300 strain containing the empty plasmid (USA300 pSD1, grey) and the NADK knockdown strain (USA300 sgRNA, blue) was monitored at OD600nm in BHI broth at 37°C (NT, open circles) or in BHI exposed to hydrogen peroxide (H2O2, closed circles). Bars indicate the standard errors of the means of biological replicates (n=3).

Figure supplement 1 shows the relQ operon for NCTC8325. Is this identical to the operon in Xen36? If so, why show for NCTC3525?

We thank the reviewer for this question. At the time of submission, the genome of the Xen36 strain had not been sequenced. Since the relQ operon is highly conserved in sequenced strains of *S. aureus*, we showed the relQ operon of the prototypical NCTC3525 strain. We agree that showing the relQ operon of Xen36 would be more relevant to our study. We have since sequenced the genome of Xen36. Figure supplement 1 (page 49) now shows the Xen36 relQ operon with corresponding coordinates.

It can be estimated that the degree of NADK knockdown is high, since the ppnK transcript and NADK protein are barely detectable in the NADK sgRNA strain (Figure supplement 1A and 1B, page 49). These results corroborate our previous findings showing that levels of NADK and NADP(H)/NAD(H) ratio are strongly reduced in the knockdoen strain (Gelin et al., 2020)

It can be estimated that the degree of NADK knockdown is high, since the ppnK transcript and NADK protein are barely detectable in the NADK sgRNA strain (Figure supplement 1A and 1B, page 49). These results corroborate our previous findings showing that levels of NADK and NADP(H)/NAD(H) ratio are strongly reduced in the knockdoen strain (Gelin et al., 2020)

Lines 122-123 fail to mention the growth defect of the KD strain, which is important and could impact many other phenotypes. Given the relationship of NADK to oxidative stress, it would be useful to see if this growth defect is abolished under anaerobic conditions. This will determine whether growth is due to increase susceptibility to oxidative stress or e.g. decreased metabolic activity.

We thank the reviewer for this comment. The growth defect attesting the efficiency of the knockdown has been introduced in the text (lines 99-100).

To test the relationship between NADK and oxidative stress, rather than studying growth under anaerobic conditions which leads to massive changes in gene expression and metabolic pathways (Fuchs et al., 2007), we measured growth of *S. aureus* containing the empty vector and NADK knockdown strains in BHI supplemented or not with catalase. As expected, growth of Xen36 in BHI was affected by NADK knockdown (Author response image 2). Addition of catalase to the culture medium decreased this growth defect (Author response image 2). These results suggest that reactive oxygen species generated during aerobic growth contribute to the growth defect of the knockdown strain.

**Author response image 2. sa2fig2:** Reactive oxygen species contribute to *S. aureus* growth defect upon NADK knockdown. Aerobic growth of the Xen36 strain containing the empty plasmid (pSD1, grey) and the NADK knockdown strain (NADK sgRNA, blue) was monitored at OD600nm in BHI broth at 37°C (NT, open circles) or in BHI supplemented with 10 ug/mL catalase (closed circles). Bars indicate the standard errors of the means of biological replicates (n=3).

Knockdown of NADK production sensitises *S. aureus* to H2O2, but it's not clear why. It would be useful to show that catalase production is unaffected in the KD strain.

The reviewer brings up an important aspect of the role of NADK, which we investigated to understand the mechanism of the increased sensitivity to H2O2 of the knockdown strain. Comparative proteomic analysis of *S. aureus*/pSD1 and *S. aureus*/NADK sgRNA strains revealed that NADK deficiency led to decreased relative abundance of *S. aureus* catalase (Table 1, page 16; Table supplement 1, page 55; Table supplement 4, page 55; Figure supplement 3, page 53). These results and our new data showing that NADK supports other major antioxidant enzymes (Table 1, page 16; Table supplement 1, page 55; Table supplement 4, page 55; Figure supplement 3, page 53), suggest that deficiency in the antioxidant arsenal upon NADK knockdown contributes to sensitize *S. aureus* to oxidative stress.

Lines 153-154: For the triclosan susceptibility experiments, the effect shown in panel 3A is not obviously different between the KD strain and control. It doesn't appear that any statistical tests were done here to show differences (the only comparisons appear to be to No Treatment, not between strains). For panel B, it's unclear why catalase promotes growth of the KD strain. Is this strain catalase deficient?

We decided to remove these experiments from the new manuscript for clarity and focus our study on the role of NADK in *S. aureus* pathogenicity and its mechanism.

The EM analysis in figure supplement 2D isn't particularly convincing since most of the cells look fine. It could be that the images represent debris from the culture. A better, more quantitative test of membrane integrity would be to use a stain such as propidium iodide. It would also be useful to examine bacterial susceptibility to agents that target the membrane such as gramicidin, nisin or daptomycin. I note the subsequent experiment with polymyxin B, but this is typically used to combat Gram-negative bacteria as it has a high affinity for LPS and it's not clear what the concentrations used in this work do to the bacterial membrane, especially since there's only ~20% growth inhibition.

For the reasons indicated above, we decided to remove these experiments from the new manuscript.

Line 189 refers to bacterial survival during antibiotic stress, but the assay measures growth. These are two very different things. Survival is usually measured by time-course killing assays and growth by determination of the minimum inhibitory concentration.

We agree with the reviewer. This sentence has been suppressed from the new manuscript.

For the antibiotic assays it is not clear to me how the antibiotics were chosen. Polymyxin B is not used to treat *S. aureus* infections, nor is kanamycin or levofloxacin used routinely. It would be informative to include an ant-staphylococcal penicillin such as oxacillin. I'm also unsure of the relevance of this assay. It seems like there are very small differences in susceptibility between KD and control strain and the drug concentrations used are chosen specifically to focus on these small differences. If the author wants to investigate the role of ROS, they should repeat experiments under anaerobic conditions.

We apologize for the lack of clarity in the text. Our goal was not to investigate a putative role of NADK in resistance to clinically relevant antibiotics. At first, we hypothesized that NADK could be involved in bacterial response to antibiotic stress. We therefore used subinhibitory concentration of antibiotics acting on different bacterial targets.

We realized that this part of the original manuscript could be source of confusion. Since antibiotic stress and pathogenicity are different aspects, we decided to remove these experiments from the new manuscript and focus on pathogenicity.

Line 194: All Gram-positive bacteria are resistant to polymyxins since they lack LPS. Activity against Gram-positives is thought to be due to the surfactant properties of the antibiotic but this is not well established. A much better choice here would be daptomycin since it is used against *S. aureus* and targets the membrane.

For the reasons indicated above, we decided to remove these experiments from the new manuscript.

When measuring survival of *S. aureus* in phagocytes, it is important to consider whether there are differences in production of cytolytic toxins, which can disrupt immune cell function. Quantitative measurements of hemolytic activity or π staining of immune cells would suffice. This is especially important since the author observed escape from phagosomes, which is likely toxin mediated.

We thank the reviewer for these helpful suggestions. We have now quantified the levels of major *S. aureus* cytolysins, hemolytic activity and immune cell viability. Strikingly, our comparative proteomic study highlights a role of NADK in toxin production (Figure 4, page 18; Table 2, page 19; Figure supplement 3, page 53; Table supplement 6, page 56). We also demonstrate that NADK promotes hemolytic activity (Figure 4, page 18) and cytotoxicity (Figure 2, page 11).

The zebrafish data could be a consequence of different growth rates of the KD and control strain or differences in toxin production. In fact, toxin production seems more likely since relatively high CFU were found in fish, but the fish survived. Furthermore, there seems to be a depletion of phagocytic cells. Therefore, it's important to determine whether there are differences in toxin production between strains. If the authors wish to confirm the importance of oxidative immune defences, they should use a fish line defective for the oxidative burst.

We thank the reviewer for these comments and suggestions. As mentioned by the reviewer, it is unlikely that different growth rates of the two strains explain the zebrafish data. Indeed, we show that similar numbers of *S. aureus*/pSD1 and *S. aureus*/NADK sgRNA are recovered from zebrafish at 6 hours and 24 hours post-infection (Figure supplement 2, page 51). We have demonstrated that NADK supports toxin production, as hypothesized by the reviewer (Figure 4, page 18; Table 2, page 19; Figure supplement 3, page 53; Table supplement 6, page 56). Additionally, we now show that production of antioxidant enzymes by the knockdown strain is defective in infected zebrafish (Table supplement 4, page 55; Table supplement 5, page 55), possibly sensitizing *S. aureus* to host defenses.

Figure 6A, B doesn't appear to have been analysed using any statistical tests.

Statistical analysis has now been performed (Figure 1A, page 8). Figure 1B showing a combination of living and dead larvae precludes a statistical test, as previously published by experts in the field of zebrafish infection (Prajsnar et al., 2008, 2012).

Line 317. ROS levels were not measured so this conclusion is unsupported. There are several other antibacterial factors in macrophages, including antimicrobial peptides that target the bacterial membrane.

This sentence has been changed. In fact, we now show that NADK protect *S. aureus* from both oxidative and cell-wall targeting compounds (Figure 3, page 14).

Line 345: If an inhibitor of NADK is available this should be used to confirm key findings.Line 392. What time point was used for antibiotic-mediated growth inhibition experiments and why?

For the reasons indicated above, we decided to remove the antibiotic experiments from the new manuscript.

Reviewer #2:This work describes the role of Staphylococcal NADK in bacterial pathogenesis. It uses CRISPR-based interference to knock-down expression of the ppnK gene that encodes for NADK in *Staphylococcus aureus*. Comparing the behaviour of wild-type and ppnK knock-down *S. aureus*, the authors show that NADK protects the bacteria against oxidative stresses directly induced by 1) peroxide, 2) inhibition of fatty acid elongation, 3) exposure to various classes of antibiotics, and 4) uptake by macrophages. In addition, they show that NADK protects against some non-oxidative stresses like osmotic shock and exposure to other antibiotics. Lastly, they show that NADK knockdown impairs *S. aureus* survival in macrophages and impairs pathogenicity in a zebrafish infection model.The observations in the paper are well described. However, more technical detail is required to unequivocally support the major conclusion.

We thank the reviewer for these comments. We have worked to add data supporting the major conclusion of our study.

Specifically, the authors start with showing that ppnK is part of an operon. Therefore, targeting ppnK via CRISPR-based interference could also affect genes downstream in this operon as described by Zhao et al. who devised the system (described in Applied and Environmental Microbiol 2017). To unequivocally show that NADK mediates the observed phenotypes, the authors should complement their knockdown strain with exogenous ppnK expression, or measure and complement NADP(H) levels.

We thank the reviewer for these helpful comments. The essential role of NADK in published experimental settings (Cassette et al., 2003; Kobayashi et al., 2003; Grose et al., 2006; Chaudhuri et al., 2009) excludes the straightforward phenotypic comparison of wild-type, deletion mutant and complemented strains. We have therefore used a CRISPR-based pSD1-mediated strategy (Zhao et al., 2017), to decrease the expression of NADK. We have previously demonstrated that our NADK sgRNA vector targets NADK as it reduces the NADP(H)/NAD(H) ratio in *S. aureus* (Gelin et al., 2020). The system designed by Zhao and colleagues is based on interference induction upon addition of ATc. As mentioned below by the reviewer, the system is leaky and reversible, making complementation experiments and their interpretation uneasy. To determine if the NADK sgRNA vector had any polar effect on the expression of the gene located downstream of ppnK in the operon, we decided to analyze the levels of rluA transcript by RT-PCR. S. aureus/pSD1 and the knockdown strain had similar levels of rluA (Figure supplement 1, page 49), while the levels of ppnK was strongly reduced in the knockdown strain (Figure supplement 1, page 49). These results suggest that targeting ppnK does not affect expression of its downstream gene.

The CRISPR-based interference that is used depends on a vector that allows for constitutive expression of sgRNA and inducible expression of dCas9 to mediate the interference. Zhao et al. described that ATc induction leads to a 100-fold knockdown of a target gene. The knockdown is reversible and non-induced leaky expression will still lead to 3-fold reduction in expression of the target gene. The authors should comment on what this means for the NADK expression during the macrophage and zebrafish infection experiments that run over longer periods. Perhaps the authors know whether initial absence of NADK during infection is sufficient to evoke the better survival of fish at 72h? Or whether the non-induced strain is also less pathogenic?

We agree with the reviewer. Non-induced basal expression of the PtetO promoter has been previously reported in vitro and in vivo (Sipo et al., 2006; Tian et al., 2009; Tatsuke et al., 2013; Akmammedov et al., 2017; Zhao et al., 2017). We made a similar observation in our experimental conditions (Gelin et al., 2020). Indeed, expression of NADK is decreased in non-induced conditions (Gelin et al., 2020). The levels of NADK expression as a function of time during infection of macrophages and zebrafish with the knockdown strain are unknown. One can hypothesize that the interference is maintained as the vector is still carried by the strain in absence of antibiotics 48 hours post-infection (Figure supplement 2A, page 51).

1. The way the introduction is written, it is not directly clear whether the paper would focus on bacterial or human NADK. Stating this earlier could make the paper easier to read.

We have now referred to bacterial NADK in the first sentence of the introduction.

2. Looking at the described experiments and the points raised in the general discussion, the regulation of NADKs by calcium and calmodulin seem redundant. Perhaps the authors can clarify why they highlight this in the introduction?

We agree with the reviewer. The paragraph on NADK regulation in plants, which was dispensable to understand the study, has been deleted.

3. Readers would benefit from a broader perspective in the discussion about known ways in which *S. aureus* deals with ROS during infection. For example, production of catalase, staphyloxanthin, and the MPO inhibitor SPIN.

We thank the reviewer for this suggestion. Since we uncovered the regulatory role of NADK on protective enzymes, toxins, and the response regulator AgrA, the discussion has been entirely rewritten to emphasize the mechanisms by which NADK promotes to *S. aureus* pathogenesis.

4. Line 111: to my opinion, this finding does not "confirm importance", but suggests it.

We agree with the reviewer. This paragraph has been deleted due to the focus of the new manuscript.

References

Akmammedov A, Geigges M, Paro R. Single vector non-leaky gene expression system for *Drosophila melanogaster*. Sci Rep. 2017 Jul 31;7(1):6899. doi: 10.1038/s41598-017-07282-w. PMID: 28761084; PMCID: PMC5537222.

Bostian PA, Karnes JM, Cui S, Robinson LJ, Daffner SD, Witt MR, Emery SE. Novel rat tail discitis model using bioluminescent *Staphylococcus aureus*. J Orthop Res. 2017 Sep;35(9):2075-2081. doi: 10.1002/jor.23497. Epub 2017 Mar 15. PMID: 27918144; PMCID: PMC5459675.

Chaudhuri RR, Allen AG, Owen PJ, Shalom G, Stone K, Harrison M, Burgis TA, Lockyer M, Garcia-Lara J, Foster SJ, Pleasance SJ, Peters SE, Maskell DJ, Charles IG. Comprehensive identification of essential *Staphylococcus aureus* genes using Transposon-Mediated Differential Hybridisation (TMDH). BMC Genomics. 2009 Jul 1;10:291. doi: 10.1186/1471-2164-10-291. PMID: 19570206; PMCID: PMC2721850.

Fuchs S, Pané-Farré J, Kohler C, Hecker M, Engelmann S. Anaerobic gene expression in *Staphylococcus aureus*. J Bacteriol. 2007 Jun;189(11):4275-89. doi: 10.1128/JB.00081-07. Epub 2007 Mar 23. PMID: 17384184; PMCID: PMC1913399.

Gelin M, Paoletti J, Nahori MA, Huteau V, Leseigneur C, Jouvion G, Dugué L, Clément D, Pons JL, Assairi L, Pochet S, Labesse G, Dussurget O. From Substrate to Fragments to Inhibitor Active *InVivo* against *Staphylococcus aureus*. ACS Infect Dis. 2020 Mar 13;6(3):422-435. doi: 10.1021/acsinfecdis.9b00368. Epub 2020 Feb 14. PMID: 32017533.

Grose JH, Joss L, Velick SF, Roth JR. Evidence that feedback inhibition of NAD kinase controls responses to oxidative stress. Proc Natl Acad Sci U S A. 2006 May 16;103(20):7601-6. doi: 10.1073/pnas.0602494103. Epub 2006 May 8. PMID: 16682646; PMCID: PMC1472491.

Kobayashi K, Ehrlich SD, Albertini A, Amati G, Andersen KK, Arnaud M, Asai K, Ashikaga S, Aymerich S, Bessieres P, Boland F, Brignell SC, Bron S, Bunai K, Chapuis J, Christiansen LC, Danchin A, Débarbouille M, Dervyn E, Deuerling E, Devine K, Devine SK, Dreesen O, Errington J, Fillinger S, Foster SJ, Fujita Y, Galizzi A, Gardan R, Eschevins C, Fukushima T, Haga K, Harwood CR, Hecker M, Hosoya D, Hullo MF, Kakeshita H, Karamata D, Kasahara Y, Kawamura F, Koga K, Koski P, Kuwana R, Imamura D, Ishimaru M, Ishikawa S, Ishio I, Le Coq D, Masson A, Mauël C, Meima R, Mellado RP, Moir A, Moriya S, Nagakawa E, Nanamiya H, Nakai S, Nygaard P, Ogura M, Ohanan T, O'Reilly M, O'Rourke M, Pragai Z, Pooley HM, Rapoport G, Rawlins JP, Rivas LA, Rivolta C, Sadaie A, Sadaie Y, Sarvas M, Sato T, Saxild HH, Scanlan E, Schumann W, Seegers JF, Sekiguchi J, Sekowska A, Séror SJ, Simon M, Stragier P, Studer R, Takamatsu H, Tanaka T, Takeuchi M, Thomaides HB, Vagner V, van Dijl JM, Watabe K, Wipat A, Yamamoto H, Yamamoto M, Yamamoto Y, Yamane K, Yata K, Yoshida K, Yoshikawa H, Zuber U, Ogasawara N. Essential *Bacillus subtilis* genes. Proc Natl Acad Sci U S A. 2003 Apr 15;100(8):4678-83. doi: 10.1073/pnas.0730515100. Epub 2003 Apr 7. PMID: 12682299; PMCID: PMC153615.

Koskinen K, Nevalainen S, Karikoski M, Hänninen A, Jalkanen S, Salmi M. VAP-1-deficient mice display defects in mucosal immunity and antimicrobial responses: implications for antiadhesive applications. J Immunol. 2007 Nov 1;179(9):6160-8. doi: 10.4049/jimmunol.179.9.6160. PMID: 17947691.

Munro P, Benchetrit M, Nahori MA, Stefani C, Clément R, Michiels JF, Landraud L, Dussurget O, Lemichez E. The *Staphylococcus aureus* epidermal cell differentiation inhibitor toxin promotes formation of infection foci in a mouse model of bacteremia. Infect Immun. 2010 Aug;78(8):3404-11. doi: 10.1128/IAI.00319-10. Epub 2010 May 17. PMID: 20479081; PMCID: PMC2916266.

Peetermans M, Vanassche T, Liesenborghs L, Claes J, Vande Velde G, Kwiecinksi J, Jin T, De Geest B, Hoylaerts MF, Lijnen RH, Verhamme P. Plasminogen activation by staphylokinase enhances local spreading of *S. aureus* in skin infections. BMC Microbiol. 2014 Dec 17;14:310. doi: 10.1186/s12866-014-0310-7. PMID: 25515118; PMCID: PMC4274676.

Planet PJ. Life After USA300: The rise and fall of a superbug. J Infect Dis. 2017 Feb 15;215(suppl_1):S71-S77. doi: 10.1093/infdis/jiw444.

Prajsnar TK, Cunliffe VT, Foster SJ, Renshaw SA. A novel vertebrate model of *Staphylococcus aureus* infection reveals phagocyte-dependent resistance of zebrafish to non-host specialized pathogens. Cell Microbiol. 2008 Nov;10(11):2312-25. doi: 10.1111/j.1462-5822.2008.01213.x. Epub 2008 Aug 18. PMID: 18715285.

Prajsnar TK, Hamilton R, Garcia-Lara J, McVicker G, Williams A, Boots M, Foster SJ, Renshaw SA. A privileged intraphagocyte niche is responsible for disseminated infection of *Staphylococcus aureus* in a zebrafish model. Cell Microbiol. 2012 Oct;14(10):1600-19. doi: 10.1111/j.1462-5822.2012.01826.x. Epub 2012 Jul 4. PMID: 22694745; PMCID: PMC3470706.

Pribaz JR, Bernthal NM, Billi F, Cho JS, Ramos RI, Guo Y, Cheung AL, Francis KP, Miller LS. Mouse model of chronic post-arthroplasty infection: noninvasive in vivo bioluminescence imaging to monitor bacterial burden for long-term study. J Orthop Res. 2012 Mar;30(3):335-40. doi: 10.1002/jor.21519. Epub 2011 Aug 11. PMID: 21837686; PMCID: PMC3217109.

Sassetti CM, Boyd DH, Rubin EJ. Genes required for mycobacterial growth defined by high density mutagenesis. Mol Microbiol. 2003 Apr;48(1):77-84. doi: 10.1046/j.1365-2958.2003.03425.x. PMID: 12657046.

Sipo I, Hurtado Picó A, Wang X, Eberle J, Petersen I, Weger S, Poller W, Fechner H. An improved Tet-On regulatable FasL-adenovirus vector system for lung cancer therapy. J Mol Med (Berl). 2006 Mar;84(3):215-25. doi: 10.1007/s00109-005-0009-1. Epub 2005 Dec 31. PMID: 16437213.

Tatsuke T, Lee JM, Kusakabe T, Iiyama K, Sezutsu H, Uchino K. Tightly controlled tetracycline-inducible transcription system for explosive gene expression in cultured silkworm cells. Arch Insect Biochem Physiol. 2013 Apr;82(4):173-82. doi: 10.1002/arch.21083. Epub 2013 Jan 31. PMID: 23371880.

Tian X, Wang G, Xu Y, Wang P, Chen S, Yang H, Gao F, Xu A, Cao F, Jin X, Manyande A, Tian Y. An improved tet-on system for gene expression in neurons delivered by a single lentiviral vector. Hum Gene Ther. 2009 Feb;20(2):113-23. doi: 10.1089/hum.2008.018. PMID: 20377365.

Trikha R, Greig D, Kelley BV, Mamouei Z, Sekimura T, Cevallos N, Olson T, Chaudry A, Magyar C, Leisman D, Stavrakis A, Yeaman MR, Bernthal NM. Inhibition of angiotensin converting enzyme impairs antistaphylococcal immune function in a preclinical model of implant infection. Front Immunol. 2020 Sep 11;11:1919. doi: 10.3389/fimmu.2020.01919. PMID: 33042111; PMCID: PMC7518049.

Zhao C, Shu X, Sun B. Construction of a gene knockdown system based on catalytically inactive ("Dead") Cas9 (dCas9) in *Staphylococcus aureus*. Appl Environ Microbiol. 2017 May 31;83(12):e00291-17. doi: 10.1128/AEM.00291-17. PMID: 28411216; PMCID: PMC5452804.